# Structural modifications in strain-engineered bilayer nickelate thin films

Lopa Bhatt[1 ✉], Edgar Abarca Morales[2], Abigail Y. Jiang[3,4], Eun Kyo Ko[5,6], Yi-Feng Zhao[7], Noah Schnitzer[8,9], Grace A. Pan[3], Dan Ferenc Segedin[3], Yidi Liu[5,10], Yijun Yu[5,6], Charles M. Brooks[3], Antia S. Botana[7], Harold Y. Hwang[5,6], Julia A. Mundy[3,4], David A. Muller[1,9] & Berit H. Goodge[2 ✉]

The discovery of high-temperature superconductivity in bulk $La_3Ni_2O_7$ under high hydrostatic pressure[1–4] and biaxial compression in epitaxial thin films[5–8] has generated substantial interest in understanding the interplay between atomic and electronic structure in these compounds. Subtle changes in the nickel–oxygen bonding environment are thought to be key drivers for stabilizing superconductivity, but specific details of which bonds and which modifications are most relevant remain unresolved so far. Although direct, atomic-scale structural characterization under hydrostatic pressure is beyond present experimental capabilities, static stabilization of strained $La_3Ni_2O_7$ films provides a platform well suited to investigation with new picometre-resolution electron microscopy methods. Here we use multislice electron ptychography (MEP)[9,10] to directly measure the atomic-scale structural evolution of $La_3Ni_2O_7$ thin films across a wide range of biaxial strains tuned by substrate choice. By resolving both the cation and oxygen sublattices, we study the strain-dependent evolution of atomic bonds, providing the opportunity to isolate and disentangle the effects of specific structural motifs for stabilizing superconductivity. We identify the lifting of crystalline symmetry through modification of the nickel–oxygen octahedral distortions under compressive strain as a key structural ingredient for superconductivity and identify in-plane lattice compression as a common attribute between bulk and thin-film superconductivity. Building on the detailed structures obtained by MEP, we introduce a theoretical framework to disentangle coupled structural distortions in corner-sharing octahedra[11], which suggest that both known superconducting geometries of $La_3Ni_2O_7$ (hydrostatic pressure and compressive strain) suppress local $t_{2g}$ orbital mixing in the low-energy Ni bands by raising the octahedral symmetry.

Attempts to identify further high-temperature oxide superconductors have earned substantial experimental and theoretical efforts since the discovery of superconductivity in cuprates. As well as square-planar nickelates[12], high-temperature superconductivity surpassing the transition temperature ($T_c$) of 80 K has recently been observed in the layered $n = 2$ Ruddlesden–Popper nickelates, $La_{n+1}Ni_nO_{3n+1}$, under high hydrostatic pressure[1–4] and $T_c$ up to 40 K under compressive epitaxial strain[5–8].

Although square-planar $R$NiO$_2$ ($R$ = hole-doped rare earth) and related $Nd_6Ni_5O_{12}$ (ref. 13) nickelate compounds seem in many ways cuprate analogues[14,15], parallels to the Ruddlesden–Popper nickelates $La_3Ni_2O_7$ (Fig. 1a) and $La_4Ni_3O_{10}$ that superconduct under pressure are less obvious. Bilayer and trilayer Ruddlesden–Popper nickelates host formal Ni configurations of $d^{7.5}$ and $d^{7.33}$ rather than cuprate-like $3d^9$ and the proposed Fermi surfaces contain prominent contributions from both the nickel $d_{z^2}$ and $d_{x^2-y^2}$ orbitals[16–22]. It has been suggested that the relevant role of different orbitals may lead to competition between different pairing symmetries from $d$-wave to $s^\pm$ (refs. 23,24). The emergence of superconductivity at low temperatures may also be linked to the concomitant suppression of a density-wave state[25]. So far, however, the relative roles of structural and electronic reconfiguration–and their interplay–for driving superconductivity in $La_3Ni_2O_7$ under pressure remain unclear. Although the high pressures required to achieve superconductivity in bulk limit accessible experimental probes, superconducting compressively strained $La_3Ni_2O_7$ thin films provide an experimental platform to isolate and disentangle specific structural or electronic contributions for superconductivity.

Here we use scanning transmission electron microscopy (STEM) to provide key insights into the atomic-scale structure of $La_3Ni_2O_7$ thin

[1]School of Applied and Engineering Physics, Cornell University, Ithaca, NY, USA. [2]Max Planck Institute for Chemical Physics of Solids, Dresden, Germany. [3]Department of Physics, Harvard University, Cambridge, MA, USA. [4]John A. Paulson School of Engineering and Applied Sciences, Harvard University, Cambridge, MA, USA. [5]Stanford Institute for Materials and Energy Sciences, SLAC National Accelerator Laboratory, Menlo Park, CA, USA. [6]Department of Applied Physics, Stanford University, Stanford, CA, USA. [7]Department of Physics, Arizona State University, Tempe, AZ, USA. [8]Department of Materials Science and Engineering, Cornell University, Ithaca, NY, USA. [9]Kavli Institute at Cornell for Nanoscale Science, Cornell University, Ithaca, NY, USA. [10]Department of Physics, Stanford University, Stanford, CA, USA. ✉e-mail: lb628@cornell.edu; berit.goodge@cpfs.mpg.de

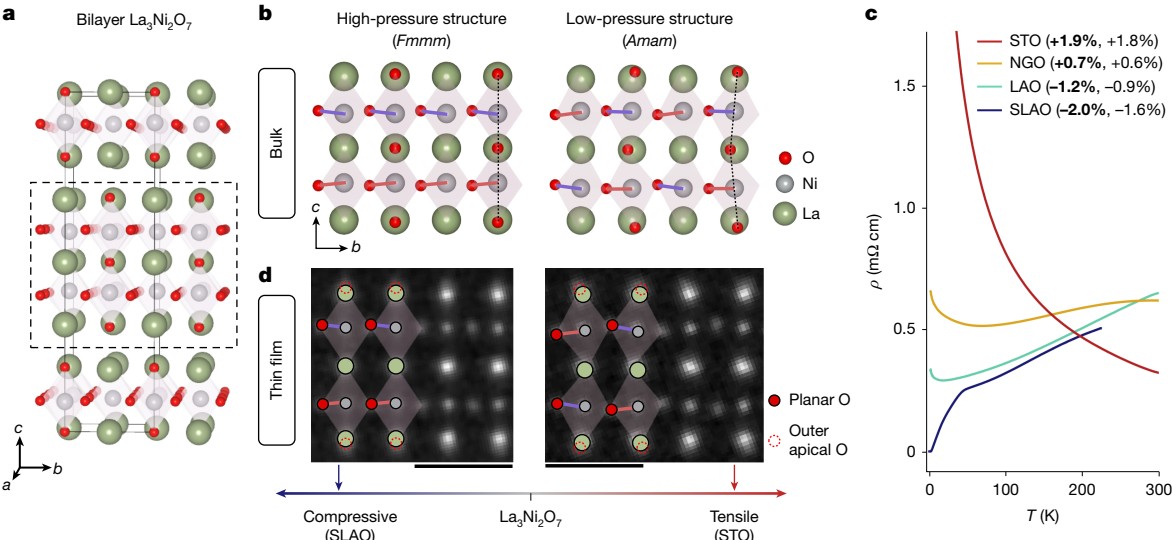

**Fig. 1 | Structural and electrical changes in La₃Ni₂O₇ thin films under compressive and tensile strain. a**, Schematic model of high-pressure *Fmmm* La₃Ni₂O₇ with the unit cell marked by a grey box. The dashed box outlines a single bilayer of La₃Ni₂O₇. **b**, High-pressure and low-pressure bulk structures of one bilayer reported in ref. 1. Dashed lines mark the Ni–apical O bond angles; Ni–planar O bond angles are highlighted by red and purple lines. **c**, Resistivity $\rho(T)$ of La₃Ni₂O₇ thin films epitaxially strained on substrates SLAO (blue), LAO (cyan), NGO (yellow) and STO (red); chemical formulas are given in the text. The nominal (bold, $\varepsilon_{nom}$) and measured epitaxial ($\varepsilon_{exp}$) strains imparted on each film are given in parentheses. The film grown on SLAO shows a superconducting onset near 42 K (ref. 5). **d**, Atomic structures of La₃Ni₂O₇ thin films under compressive (left) and tensile (right) strain measured experimentally by MEP with coloured circles identifying the La (green), Ni (grey), planar O (filled red) and outer apical O (open dashed red) atomic columns. Scale bars, 5 Å.

films with varying epitaxial strain. Using high-resolution STEM imaging and MEP[9,10], we directly characterize structural parameters including octahedral Ni–O bond angles, bond lengths and lattice constants across a complete strain series to isolate common motifs in superconducting La₃Ni₂O₇. Our measurements reveal that the Ni–planar O bond coordination takes on specific configurations of higher (lower) symmetry under compressive (tensile) strain, pointing to the importance of raising crystalline symmetry for superconductivity in these materials. Compared with bulk crystals under hydrostatic pressure, we observe a distinct evolution of out-of-plane Ni–O bond lengths but a common trend of in-plane bond lengths in compressively strained thin films. With quantitative parameterization of atomic positions and lattice constants across the full strain series, these measurements provide accurate structural models for ab initio electronic structure calculations. We introduce a theoretical framework to disentangle coupled structural distortions in corner-sharing octahedra (for example, bond length changes, rotations) and their impact on electronic structure in model systems. This analysis reveals that both known superconducting geometries of La₃Ni₂O₇ under hydrostatic pressure and compressive strain are characterized by symmetry-driven expulsion of $t_{2g}$ orbital mixing in low-energy Ni bands. Our detailed structural study of La₃Ni₂O₇ thin films across a range of compressive and tensile strain sheds light on the complex relation between structural and electronic degrees of freedom in this new class of high-temperature superconductors.

## Electrical properties of thin films

Epitaxially strained thin films of La₃Ni₂O₇₋δ are synthesized on substrates SrLaAlO₄ (SLAO), LaAlO₃ (LAO), NdGaO₃ (NGO) and SrTiO₃ (STO) to impart nominal epitaxial strains ($\varepsilon_{nom}$) of −2.0% (SLAO), −1.2% (LAO), +0.7% (NGO) and +1.9% (STO) (Methods). Experimentally measured biaxial strain values ($\varepsilon_{exp}$) show partial lattice relaxation in all films (Fig. 1c and Supplementary Table 1). Here negative (positive) values denote compressive (tensile) strain. For consistency, we use the lattice convention shown in Fig. 1a, in which $c$ is the long axis along the layer stacking direction and refer to $\varepsilon_{nom}$ throughout.

Electronically, the film under high compressive strain on SLAO shows a broad superconducting transition at ambient pressure with onset close to 42 K, reaching zero resistance around 2 K (ref. 5) (Fig. 1c). Films grown on LAO and NGO show metallic behaviour, with LAO being more metallic than NGO. Films on STO are insulating, in agreement with previous studies[5,26].

So far, superconductivity has been demonstrated in strained bilayer nickelate thin films grown by pulsed laser deposition (PLD)[5,27–32], molecular beam epitaxy (MBE)[7,33,34] and gigantic oxidative atomic layer-by-layer epitaxy[6]. In this work, the films on STO, NGO and LAO are grown by MBE, whereas the superconducting film on SLAO is grown by PLD. Although we cannot definitively exclude the possibility that properties such as superconductivity may be sensitive to the growth method or other details of film synthesis, comparison between electronic transport and atomic structure of MBE-grown and PLD-grown films on LAO finds good agreement between both techniques (Extended Data Fig. 1). These observations are in line with angle-resolved photoemission experiments, which found no distinction between the electronic structure measured in MBE-grown or PLD-grown superconducting films[33].

## Strain-dependent Ni–O bond symmetry

The compression-induced straightening of the Ni–apical O bond angle in bulk La₃Ni₂O₇ (Fig. 1b) thought to modify Ni–O hybridization and Fermi surface structure correlates to a symmetry raising from orthorhombic *Amam* to an *Fmmm* or tetragonal *I4/mmm* structure[1,35,36]. The variability of crystal structures and symmetries reported so far highlights the experimental difficulty in distinguishing similar phases—especially in the presence of mixed structural domains or twins[3,35–37]—and emphasizes a need for direct, real-space measurements of octahedral structure (Supplementary Figs. 12 and 13).

Quantitative measurements of light element positions—especially in the presence of heavy ions—with high spatial resolution and precision is, however, experimentally challenging. Local cross-sectional imaging in the STEM provides a platform for directly measuring atomic structure in thin films but particular care must be taken for robust

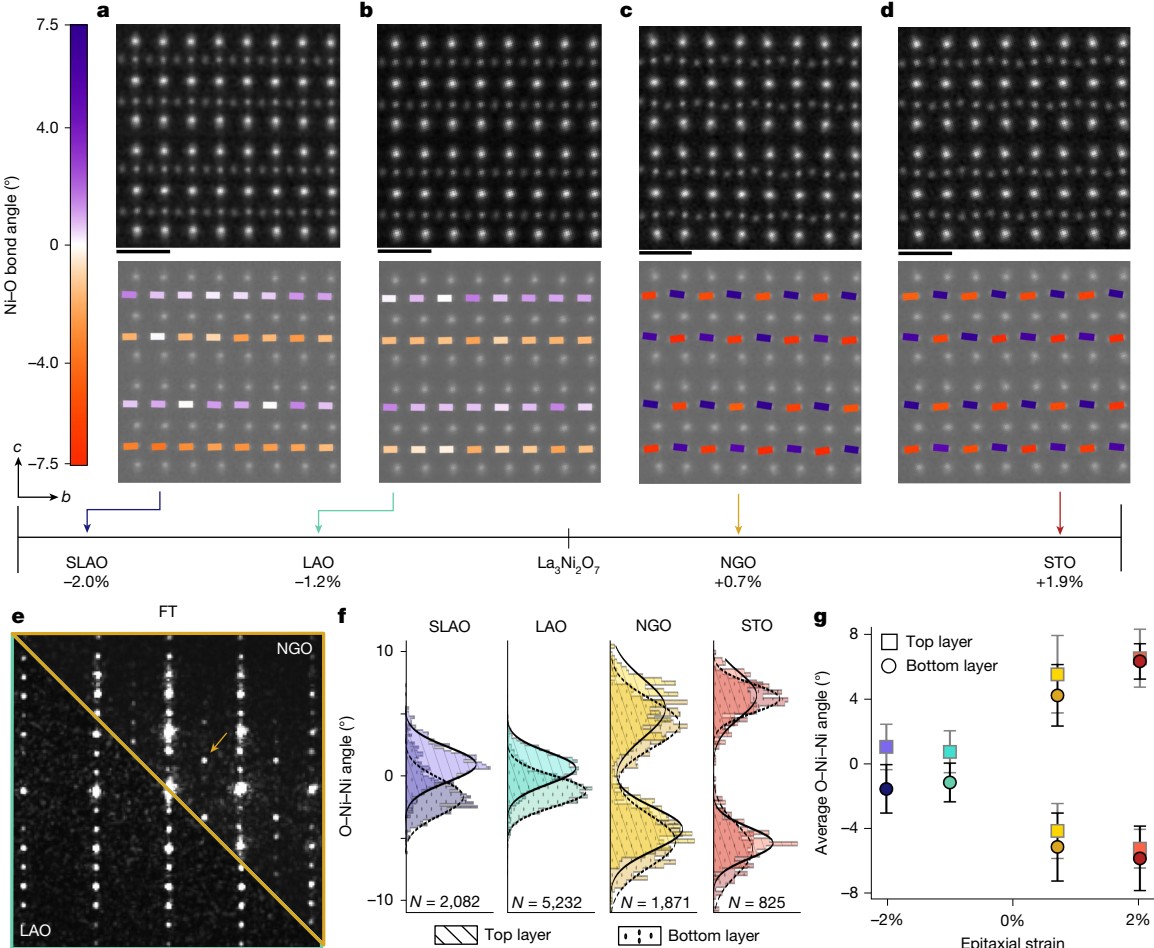

**Fig. 2 | Ni–O bond angle evolution across epitaxial strain in La₃Ni₂O₇ thin films. a–d**, MEP reconstructions (top) and Ni–planar O bond angle maps (bottom) of La₃Ni₂O₇ thin films grown on SLAO, LAO, NGO and STO substrates. **e**, Fourier transforms (FT) of ADF images of thin films on NGO (upper right) and LAO (bottom left). Differences in symmetry between tensile and compressively strained films are apparent by additional superlattice peaks that indicate a lower crystalline symmetry. **f**, Ni–planar O (O–Ni–Ni) angles of all four thin films. The solid line is a Gaussian fit to the top layer angle distribution and the dashed line is a Gaussian fit to the bottom layer angle distribution. The total number of O–Ni–Ni angles analysed (*N*) are given for each film. **g**, Average Ni–planar O angle of the top and bottom layers for films under different epitaxial strains. Scale bars, 5 Å.

measurements of light elements such as oxygen. Although certain collection geometries such as annular bright field or integrated differential phase contrast can resolve light element positions in carefully prepared specimens, they are susceptible to artefacts based on the experimental set-up (such as crystalline projection alignment, aberrations in the imaging probe) and sample geometry (such as specimen thickness, electron dechannelling near defects), which hinder quantitative measurements[38,39].

MEP is a recently implemented imaging technique enabling quantitative measurement of light atoms while eliminating many such artefacts. By solving the inverse problem of electron scattering to attain the sample potential, MEP provides deep sub-angstrom spatial resolution, depth-resolved information along the projection direction and interpretable contrast of both light and heavy atomic columns[9] (Supplementary Figs. 1 and 2). MEP therefore offers a unique capability to quantify subtle structural distortions in NiO₆ octahedra across the series of strained La₃Ni₂O₇ thin films.

Figure 1d shows MEP reconstructions of compressive and tensile strained La₃Ni₂O₇ thin films projected along the orthorhombic *a*-axis ([100]ₒ). Lanthanum, nickel and planar oxygen atomic columns are clearly visible with high spatial resolution. Apical O are visible as faint tails of outer La columns, as indicated by dotted red circles (Fig. 1d). Qualitatively, the tails are aligned vertically with Ni columns in compressive films but tilted to alternating sides of the La sites in tensile

strained films. This suggests that the Ni–apical O bond angles (dashed lines) are closer to 180° in compressive films than in tensile films. A similar trend is observed in density functional theory (DFT)-based structural relaxations[40,41]. Robust experimental quantification of precise vertical bond angles is, however, problematic owing to the projection nature of STEM imaging: although signatures of the O columns are clearly discernible in the orthorhombic projection shown here, their overlap with La columns makes precise measurement of atomic positions challenging. O and La columns are easily distinguished along the [110]ₒ projection, but in this orientation, octahedral rotations in the low-symmetry space group are antiphase stacked, which results in an apparent ellipticity of O sites in incoherently distorted octahedra[10] (Supplementary Fig. 1).

We instead track the angle of distortion between Ni and planar O (red and purple lines in Fig. 1b,c), which also rearrange in response to pressure to fingerprint structural changes in thin films across varying strain. Here we focus on the projected octahedral distortion by measuring the Ni–planar O angle with respect to the horizontal Ni plane as illustrated in Supplementary Fig. 11, giving the O–Ni–Ni angle. Related Ni–O–Ni angles are illustrated and given in Supplementary Figs. 11 and 14.

Figure 2a–d, top shows MEP reconstructions of La₃Ni₂O₇ thin films spanning epitaxial strain from approximately −2% to +2%, each cropped to two bilayers for display purposes but representing the overall structure of each film (Methods and Supplementary Fig. 3).

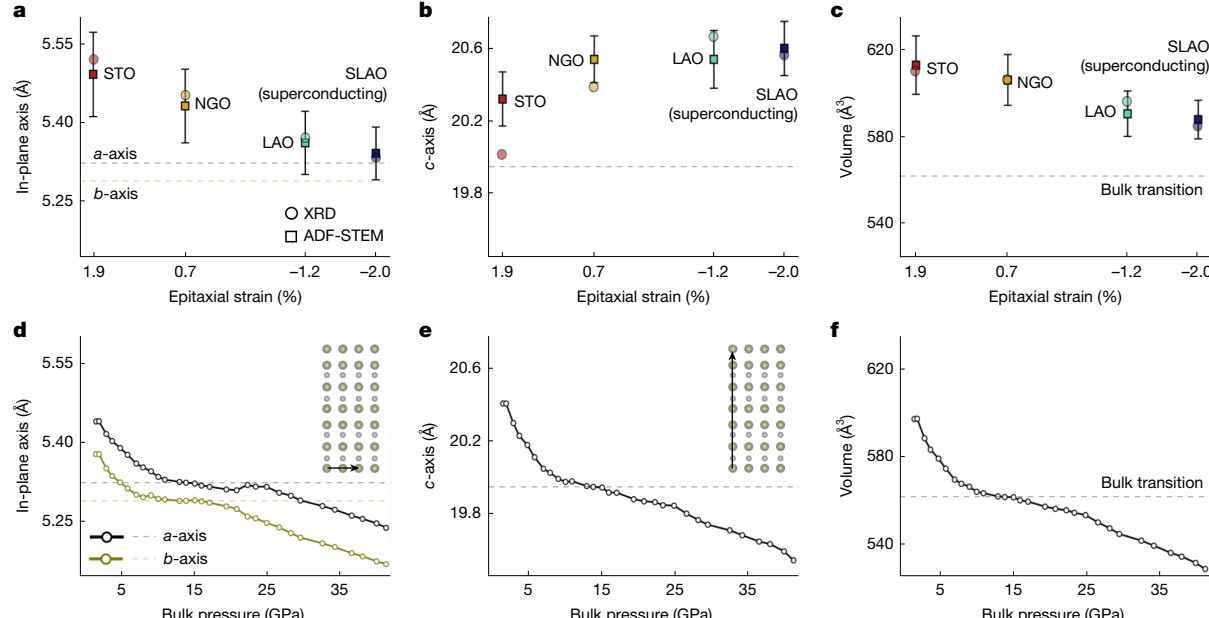

**Fig. 3 | Lattice parameter changes across epitaxial strain in La₃Ni₂O₇ thin films. a–c,** In-plane lattice constant (**a**), *c*-axis lattice constant (**b**) and unit cell volume (**c**) of thin films as a function of epitaxial strain measured using XRD (circles) and ADF-STEM (squares). Dashed lines indicate the reported values of each parameter at the onset of superconductivity and structural transitions in bulk La₃Ni₂O₇. **d–f,** In-plane lattice constant (**d**), *c*-axis lattice constant (**e**) and unit cell volume (**f**) versus pressure in bulk La₃Ni₂O₇ from ref. 1.

Maps of the individual bond angles overlaid on the MEP reconstructions are presented in the bottom row. Most notable is the qualitative difference between the symmetry of compressive and tensile strained films: across a single NiO₂ plane in each bilayer, Ni–planar O angles are aligned unidirectionally in compressive films but alternate in tensile films, as also clearly exhibited in O–Ni–Ni angle histograms separated between alternating sites (Supplementary Fig. 15). This change in symmetry is further visible through the presence of extra peaks in Fourier transforms of annular dark field (ADF)-STEM images of each sample (Fig. 2e and Supplementary Fig. 8). Further half-order peaks visible in tensile strained films (orange arrow in Fig. 2e) and lack thereof in compressive films indicate relatively lower and higher crystalline symmetries, respectively, in good agreement with expected patterns for structures reported for La₃Ni₂O₇ under low-pressure and high-pressure conditions[1,35,40]. Theoretical calculations also predict compressive stabilization of a higher-symmetry structure under compressive strain[40].

The octahedral distortions we observe in compressively strained films show a close match to the originally reported structure of bulk La₃Ni₂O₇ under high pressure at ambient temperatures[1], in which planar O shift unidirectionally away from the centre LaO plane towards the rock salt layer (Supplementary Fig. 13). This is distinct from the uniformly inward distortions reported[35] under high-pressure, low-temperature conditions (Supplementary Fig. 13). Whether superconducting thin films undergo a similar low-temperature inversion of O distortion direction remains an open question.

As well as this change in symmetry, MEP measurements also reveal a systematic trend in the magnitude of in-plane O–Ni–Ni angles across the strain series. Measuring several MEP reconstructions across each film, we find that the average O–Ni–Ni angle increases with increasing strain in both tensile and compressive directions (Fig. 2f,g). Specifically, superconducting film on SLAO has an average O–Ni–Ni angle larger than the metallic film on LAO with similar octahedral symmetry but smaller than high-pressure bulk La₃Ni₂O₇ refinements[1,35] (Supplementary Fig. 13). These trends are reproduced by DFT relaxations of La₃Ni₂O₇ biaxially strained to SLAO and STO lattice constants (Supplementary Tables 2 and 3).

## Lattice evolution across strain series

As well as understanding subtle changes in the structure of Ni–O bonds on compression, we also analyse in-plane and out-of-plane lattice constants examined both globally by X-ray diffraction (XRD) and locally at the atomic scale by ADF-STEM imaging. In-plane lattice constants measured using STEM and XRD decrease as expected with increasing compressive strain, similar to the decreasing trend measured in bulk crystals under hydrostatic pressure (Fig. 3a,d). The in-plane spacing of the superconducting film on SLAO closely matches that of bulk samples at the symmetry-raising structural transition and onset of superconductivity.

Notably, this trend inverts for the out-of-plane lattice constant in thin films (Fig. 3b and Supplementary Fig. 17), in which the Poisson effect drives an expansion of the *c*-axis spacing under compression such that the superconducting film on SLAO has an out-of-plane lattice spacing furthest from bulk samples under high pressure (Fig. 3e). MEP further enables precise measurement of out-of-plane Ni–O bond lengths, which is more directly related to the corresponding orbital overlap. Extended Data Fig. 2 shows out-of-plane Ni–O spacing measured from MEP reconstructions along the [100]ₚc projection, distinguished by Ni to inner apical O (circles) and Ni to outer apical O (squares). A clear increasing trend in the out-of-plane Ni–O spacing is observed, consistent with *c*-axis expansion with increasing compressive strain. The ratio between the two bond lengths (Ni–outer apical O/Ni–inner apical O) remains relatively constant across the strain series. The unit cell volume of the strained films (Fig. 3c) decreases with compression as expected, although even at the highest compression in the superconducting film on SLAO, the volume is much larger than superconducting bulk samples under hydrostatic pressure (Fig. 3f), also in agreement with theoretical relaxations (Supplementary Tables 1–3).

So far, great emphasis has been given to compression of the *c*-axis in relation to superconductivity under pressure and related emergence of an extra Ni $d_{z^2}$ state near the Fermi level[1,17,21,22]. Our results, however, clearly show an increase in *c*-axis—and in particular out-of-plane Ni–O bond lengths—in the superconducting film,

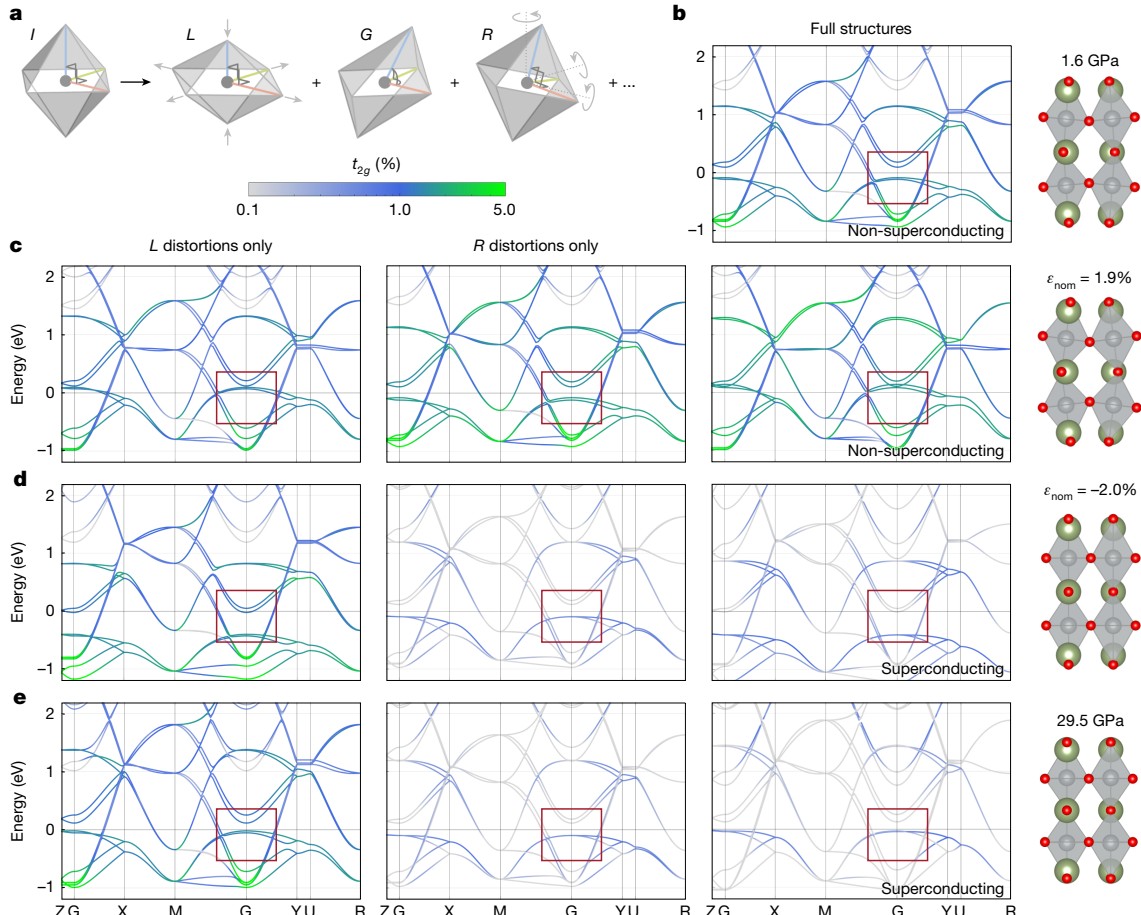

**Fig. 4 | Octahedral-distortion-decomposed electronic structure calculations. a**, Schematic representation of different distortions applied to a regular octahedron (*I*), including arm length changes (*L*), internal angle changes (*G*) and rigid rotations (*R*). The deformation of a system of corner-sharing octahedra can be decomposed into structures embodying the independent distortions. **b**–**e**, Low-energy band structures of La$_3$Ni$_2$O$_7$ projected to the local $t_{2g}$ manifold of Ni 3$d$ states. Red boxes highlight the low-energy bands near Γ discussed in the text. **b**, 1.6-GPa bulk structure at

ambient temperature reported in ref. 1. **c**–**e**, Calculated band structures for atomic models with only *L* (leftmost column) and *R* (middle column) distortions obtained from decomposing the strain between the 1.6-GPa bulk La$_3$Ni$_2$O$_7$ in **b** and the structures for epitaxial tensile strain of 1.9% on STO (**c**); epitaxial compressive strain of −2.0% on SLAO (**d**); and the 29.5-GPa bulk structure at ambient temperature reported in ref. 1 (**e**). Band calculations for the full structures are in the rightmost column. Atomic models are depicted to the right of each row.

indicating that *c*-axis compression is not necessary to achieve super-conductivity in La$_3$Ni$_2$O$_7$ thin films, thereby raising questions about relevant electronic bands. Hopping parameters calculated from our structural models of compressive (SLAO) and tensile (STO) strained films as well as for previously published structures of bulk phases under low-pressure and high-pressure conditions (Extended Data Tables 1 and 2) indicate that the ratio of interlayer $d_{z^2}$ hopping $t_\perp^z$ to intralayer $d_{x^2-y^2}$ hopping $t_\parallel^x$ (which has been previously dis-cussed in the context of pairing symmetry[21,23,41]) decreases with both compressive strain in thin films as well as hydrostatic pressure in bulk, despite opposite evolutions of their out-of-plane bond lengths.

The combined parametrization of lattice structures in strained films and pressurized bulk crystals indicates two primary structural motifs common between the two superconducting geometries: in-plane lat-tice compression and higher octahedral symmetry, here with uniform outward Ni–O plane buckling. Experimentally, separating these con-tributions is challenging or impossible, as macroscopic manipulation of one (that is, hydrostatic pressure or strain) will induce a coupled modification to the other (lattice distortions). Instead, we develop and apply a DFT-based strain-decomposition method to isolate the effects of these structural motifs on the electronic bands systematically across different sample geometries.

## Octahedral decomposition

To qualitatively disentangle the impact of structural parameters charac-terized in Figs. 2 and 3, we use a generalized theoretical framework for modelling distortions in two-dimensional systems of corner-sharing octahedra[11] (Supplementary Information Section IX). This model decomposes the total strain applied to corner-sharing octahedra into partial strain tensors corresponding to specific distortions: octahedral arm length changes (*L*), internal angle changes (*G*), rigid rotations (*R*) and their mixing into higher-order terms (Fig. 4a, Extended Data Fig. 3 and Supplementary Information Section IX). From these component matrices, we generate model octahedral geometries that embody each specific distortion to track their individual effects on the electronic structure in isolation.

With this framework, we qualitatively study the evolution of elec-tronic structure in La$_3$Ni$_2$O$_7$ across varying biaxial strain conditions and pressure. Taking the non-superconducting ambient-temperature structure of bulk La$_3$Ni$_2$O$_7$ at 1.6 GPa reported in ref. 1 (Fig. 4b) as refer-ence, we examine three more geometries: (1) the non-superconducting La$_3$Ni$_2$O$_7$ film with $\varepsilon_{nom}$ = 1.9% (Fig. 4c); (2) the superconducting film with $\varepsilon_{nom}$ = −2.0% (Fig. 4d); (3) the superconducting bulk 29.5-GPa high-pressure structure at ambient temperature (Fig. 4e) (Methods and Supplementary Information Section IX). Atomic models and projected

band-structure calculations for each of the considered structures are shown in the rightmost columns of Fig. 4, reproducing similar evolution reported previously[1,42,43]. Band structures for individual $L$ and $R$ distortions are shown in the left and middle columns; expanded energy ranges and $G$-distortion components are shown in Supplementary Fig. 21. Although the individual $L$, $G$ and $R$ structures are hypothetical, those in the rightmost column are, by definition, the DFT-relaxed structures of each sample geometry (Supplementary Table 3).

A clear distinction between the electronic structures of non-superconducting and superconducting sample geometries is observed on colouring the bands by fractional local $t_{2g}$ orbital character. Although the low-energy bands (red boxes in Fig. 4) are dominantly $e_g$-like, we find small but non-zero $t_{2g}$ hybridization in both non-superconducting bulk and tensile strained thin-film structures (Fig. 4b,c). Residual $t_{2g}$ hybridization is considerably suppressed in the two nominally superconducting −2.0% compressive strain thin-film and high-pressure bulk structures (Fig. 4d,e). The importance of local $t_{2g}$ expulsion for superconductivity remains to be established, although previous work has shown a sensitive dependence of the leading superconducting instability symmetry on the Ni crystal field splitting[23].

Our strain decomposition reveals that '$t_{2g}$-cleaning' in low-energy bands is mainly driven by $R$ distortions—that is, by the change in octahedral rotations—in combination with electron correlations phenomenologically considered in the Hubbard $U$ (Supplementary Figs. 21 and 22). The role of octahedral rotation in modifying $t_{2g}$-derived and $e_g$-derived band hybridization for superconductivity has previously been discussed for Ruddlesden–Popper oxides $Sr_2RuO_4$ and $Sr_3Ru_2O_7$ (refs. 44,45). So far, these effects have been largely unexplored in the $La_3Ni_2O_7$ literature and warrant further consideration.

Bond length ($L$) and internal angle ($G$) distortion changes, on the other hand, preserve or even enhance few-percent $t_{2g}$ mixing in all structures (Fig. 4 and Supplementary Fig. 21). Comparing calculated band structures for full atomic models and those with $L$ distortions only, we further identify that bond length changes predominantly drive energy shifts, whereas the energy levels of $G$ and $R$ distortion structures change only slightly from the near-ambient 1.6-GPa reference structure (Extended Data Figs. 4 and 5). This observation further highlights the need for extremely high-precision measurements of experimental atomic structures, as subtle differences in Ni−O bond lengths can greatly affect the position of low-energy bands. These considerations are particularly relevant for the interpretation of recent photoemission experiments[33,46], in which discrepancies in band positions could arise from structural relaxation near defects or at the film surfaces.

## Secondary phases and defects

Although STEM techniques cannot directly identify superconducting regions within the film, high-quality thin films[7,29–31] have been found to exhibit predominantly bilayer structure and demonstrate bulk-like superconductivity[31], suggesting that the pristine bilayer regions examined in Figs. 2 and 3 and used for DFT calculations probably host superconductivity. Still, comprehensive local characterization of other existing inhomogeneities including strain relaxation and crystalline defects is crucial for understanding sources of further contributions to bulk-sensitive measurements (for example, electronic transport, X-ray techniques), as well as for synthesis optimization efforts. Large-field-of-view ADF-STEM images of the superconducting $La_3Ni_2O_7$ thin film on SLAO reveal a variety of defects and relaxation mechanisms, including strain-relieving dislocations, surface relaxation, step edges in the substrate leading to extended defects in the film and mixed Ruddlesden–Popper phases (Extended Data Fig. 6 and Supplementary Information Section III). Furthermore, investigation of oxygen occupancy—essential for superconductivity in bulk[47–49] and thin films[5,6]—at mesoscopic length scales (Extended Data Fig. 6) by electron

energy loss spectroscopy at the O K-edge shows local variability in oxygen stoichiometry within a single $La_3Ni_2O_7$ sample[10] (Supplementary Information Section III). It is likely that concentrated oxygen vacancies and proximity to defects may also modify the local octahedral coordination, as seen in Extended Data Fig. 6e. Together, these observations highlight the tremendous opportunity for improving $La_3Ni_2O_7$ thin-film quality with the hopes of achieving sharper and higher-temperature superconducting transitions[7,8,28,31].

## Conclusion and outlook

Through a systematic study of $La_3Ni_2O_7$ thin films grown on different substrates with varying epitaxial strain, we identify essential structural characteristics and their effect on electronic changes across the strain series. We find commonality between superconducting strained thin films on SLAO and bulk crystals under hydrostatic pressure in the compression of in-plane lattice constants. On the other hand, opposite evolution in out-of-plane Ni−O bond lengths between these geometries result in dissimilar positioning of the Ni $d_{z^2}$ band near Γ. Direct quantitative measurements of Ni−O bond angles using MEP reveal a lifting of crystalline symmetry under biaxial compressive strain, which mirrors that reported in bulk crystals under high-pressure conditions. Our method for structural analysis by geometric decomposition in strained octahedral systems enables us to decouple the effects of experimentally measured atomic distortions on electronic band structure, suggesting the possible importance of octahedral symmetry for eliminating $t_{2g}$ orbital hybridization in low-energy nickel bands in superconducting thin-film and bulk structures. The combined experimental and theoretical approaches here demonstrate the potential for high-precision analysis of subtle structural distortions and their impact on materials properties, generalizable across a wide range of functional materials.

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

# Methods

## Thin-film synthesis

Substrates SLAO, LAO, NGO and STO have pseudo-cubic in-plane lattice constants of 3.756 Å, 3.787 Å, 3.858 Å and 3.905 Å, respectively. Following $\varepsilon_{nom} = (a_{sub} - a_{bulk})/a_{bulk}$, the changes of the in-plane lattice constant compared with ambient-temperature, ambient-pressure bulk $La_3Ni_2O_7$ (pseudo-tetragonal $a_{pt} = \sqrt{a^2 + b^2}/2 = 3.833$ Å (ref. 5)) impart nominal epitaxial strains ($\varepsilon_{nom}$) of −2.0% (SLAO), −1.2% (LAO), +0.7% (NGO) and +1.9% (STO).

The preparation of $La_3Ni_2O_7$ thin films on SLAO(001) substrates with PLD is detailed in ref. 5. Epitaxial thin films of $La_3Ni_2O_7$ on LAO(100), NGO(110) and STO(001) were grown by means of ozone-assisted MBE following the preparation in refs. 13,50, with improved structural quality achieved by shuttering LaO and $NiO_2$ monolayers according to ref. 51. The MBE-grown films were annealed in an OTF-1200X-S vacuum tube furnace (MTI Corporation) under 1 atm $O_2$ pressure for 3 h at 500 °C. All films are treated with post-growth annealing in molecular oxygen or ozone with the goal of obtaining oxygen occupancy close to stoichiometric $La_3Ni_2O_7$, although we note that the precise quantification of oxygen stoichiometry in both thin films and bulk crystals is experimentally challenging[10].

## Bulk characterization

The structural and electronic characterization of $La_3Ni_2O_7$ thin films on SLAO(001) substrates are detailed in ref. 5. For the thin films of $La_3Ni_2O_7$ on LAO(100), NGO(110) and STO(001), XRD was collected on a Malvern Panalytical Empyrean diffractometer with Cu K$\alpha_1$ radiation ($\lambda = 1.5406$ Å). $c$-axis lattice parameters were calculated with Nelson–Riley fitting[52] and in-plane parameters were obtained from the film peak centroid in reciprocal space maps collected with a PIXcel3D area detector. Electrical transport data were collected on a Quantum Design Physical Property Measurement System with ultrasonically wire-bonded aluminium–silicon contacts.

## Electron microscopy

Cross-sectional STEM specimens were prepared by the standard focused ion beam lift-out method on a Thermo Fisher Scientific G4-UX FIB to projection thicknesses of <30 nm. For the experiments performed in this manuscript, we have taken extensive precautions to mitigate oxygen loss during our experiments and to try to account for any potential effect on the structural measurements reported here. The timing of our experiments is carefully coordinated such that all focused ion beam preparation and STEM measurements are performed within three days of oxygen annealing. By comparison, when measurements are not as carefully time-coordinated, we observe a complete absence of the O K-edge pre-peak across the sample and inconsistent non-uniform Ni–O angles, shown in Supplementary Fig. 16.

ADF-STEM images were acquired using a Cs-corrected Thermo Fisher Scientific Spectra 300 X-CFEG operating at 300 kV with a probe convergence angle of 30 mrad. Ptychographic 4D-STEM datasets were collected on the same tool equipped with an EMPAD-G2 detector[53]. The maximum likelihood algorithm implemented in the fold-slice package was used to perform iterative phase retrieval[9,54,55], using position correction and multiple probe modes to account for partial coherence of the STEM probe[56,57]. A Bayesian optimization algorithm with data error as the objective function was used to optimize parameters such as defocus, convergence angle and rotation angle[58]. The typical ranges of optimized parameter values used in MEP reconstructions are given in Supplementary Tables 4 and 5.

The MEP bond angle and bond length analyses in this manuscript are based on 40 distinct datasets, with each dataset spanning a unique field of view of 10 × 10 nm, corresponding to a total of 2,082 (SLAO), 5,232 (LAO), 1,871 (NGO) and 825 (STO) bonds analysed, as indicated in Fig. 2f. ADF-STEM analysis of lattice spacing includes 23 unique datasets, each spanning a total of 30 × 30 nm field of view. All of the structural coordination analysis is based on atomic column positions determined by seven-parameter Gaussian fitting.

Electron energy loss spectroscopy at the O K-edge was measured on a second Thermo Fisher Scientific Spectra 300 X-CFEG equipped with a Gatan Continuum spectrometer and camera operating at 120 kV accelerating voltage. Probe currents were limited to <20 pA to reduce possible beam damage[10,59].

## First-principles calculations

We used DFT calculations as implemented in the Vienna Ab initio Simulation Package (VASP) to perform structural optimizations in $La_3Ni_2O_7$ under different strain levels (applied to both the *Amam* and *Fmmm* structures). To that end, we constrained the in-plane lattice constants to the appropriate biaxial strain level $\epsilon = \frac{a^*(b^*)}{a(b)}$ and relaxed both the out-of-plane lattice constant and the internal coordinates. The exchange-correlation functional chosen was the Perdew–Burke–Ernzerhof version of the generalized gradient approximation. An energy cut-off of 500 eV and a $k$-mesh of 8 × 8 × 3 were used for the different strain levels. We fully optimized both the internal coordinates and the $c$-axis until the Hellman–Feynman forces were lower than 1 meV Å$^{-1}$. Crystal models are visualized using VESTA[60].

The structure of non-superconducting $La_3Ni_2O_7$ thin film with $\varepsilon_{nom} = 1.9\%$ was obtained by relaxing the internal coordinates provided in ref. 1 of bulk 1.6-GPa structure at ambient temperature with the in-plane lattice parameters of STO (Fig. 4c). The superconducting film structure with $\varepsilon_{nom} = -2.0\%$ was obtained by relaxing the internal coordinates of bulk 29.5-GPa high-pressure structure at ambient temperature from ref. 1 with the in-plane lattice parameters of SLAO (Fig. 4d). Supplementary Table 3 tabulates structural parameters for strained $La_3Ni_2O_7$ calculated from our DFT relaxations and projected for direct analogy to STEM measurements.

## Octahedral distortion decomposition and calculations

The effect of crystalline distortions in strained $La_3Ni_2O_7$ was analysed using the 4-octahedra model introduced in ref. 11. To apply this framework to $La_3Ni_2O_7$, the model was first generalized to account for non-collinear apical distortions, which are present in this material (Supplementary Information Section IX). Subsequently, the octahedral geometry of the initial and final $La_3Ni_2O_7$ unit cells was examined. The pair of two-dimensional layers forming each bilayer in the unit cell were independently parametrized, decomposed and reassembled to construct the $L$, $G$ and $R$ partial structures.

The $La_3Ni_2O_7$ partial structures were optimized in VASP using the same parameters described above but with the unit cell and octahedral geometry (that is, O positions) held fixed, whereas the La and Ni atomic positions were allowed to relax (Supplementary Information). Band-structure calculations were performed in Quantum ESPRESSO v6.4.1. The exchange-correlation functional used was the Perdew–Burke–Ernzerhof formulation of the generalized gradient approximation, with and without an on-site Hubbard $U = 4$ eV applied to the Ni sites (Supplementary Figs. 21 and 22). An energy cut-off of 60 Ry and a 5 × 5 × 2 $k$-mesh were used.

## Data availability

The experimental datasets and crystal structures used in this study are available from Zenodo at https://doi.org/10.5281/zenodo.18917988 (ref. 61). The source data for Figs. 2g and 3a–c are provided in Supplementary Information Section I. Further data and analysis that support the conclusions of this work are available on request to the authors.

## Code availability

The codes for the ptychographic reconstructions are available from Zenodo at https://doi.org/10.5281/zenodo.13787851 (ref. 62). Scripts for the octahedral decomposition are available at https://github.com/AMoralesEdgar/The-4-Octahedra-Model.

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

**Acknowledgements** L.B. dedicates this paper to L. F. Kourkoutis for her continuous guidance, support and inspiration. We thank M. R. Norman, B. Y. Wang and Y. Tarn for useful feedback. This work made use of the Cornell Center for Materials Research shared instrumentation facility. L.B. and D.A.M. acknowledge support by the National Science Foundation (NSF) Platform for the Accelerated Realization, Analysis, and Discovery of Interface Materials (PARADIM) under cooperative agreement no. DMR-2039380. The Thermo Fisher Scientific Spectra 300 X-CFEG was acquired with support from PARADIM, NSF Materials Innovation Platforms (MIP; DMR-2039380) and Cornell University. A.Y.J., G.A.P., D.F.S., C.M.B. and J.A.M. acknowledge support from the U.S. Department of Energy, Office of Basic Energy Sciences, Division of Materials Sciences and Engineering, under award no. DE-SC0021925. A.Y.J., G.A.P. and D.F.S. acknowledge support from the NSF Graduate Research Fellowship. G.A.P. and A.Y.J. were also supported by the Paul & Daisy Soros Fellowships for New Americans and A.Y.J. by the Ford Foundation. J.A.M. acknowledges support from a Packard Fellowship and a Sloan Fellowship. E.K.K., Y.Y., Y.L. and H.Y.H. acknowledge support from the U.S. Department of Energy, Office of Basic Energy Sciences, Division of Materials Sciences and Engineering (contract no. DE-AC02-76SF00515) and the Gordon and Betty Moore Foundation's Emergent Phenomena in Quantum Systems Initiative (grant no. GBMF9072, synthesis equipment). Part of the sample fabrication was conducted at the Stanford Nano Shared Facilities (SNSF), supported by the NSF under grant ECCS-1542152. Y.-F.Z. acknowledges support from NSF grant no. DMR-2323971. A.S.B. was supported by the Alfred P. Sloan Foundation (FG-2022-19086). E.A.M. and B.H.G. were supported by the Max Planck Society. B.H.G. was also supported by Schmidt Science Fellows in partnership with the Rhodes Trust.

**Author contributions** The project was conceived by L.B., B.H.G. and J.A.M. Thin films on LAO, NGO and STO were synthesized by MBE by A.Y.J., G.A.P. and D.F.S., with assistance from C.M.B. and J.A.M. Thin films on SLAO and LAO were synthesized by PLD by E.K.K., Y.L. and Y.Y., with supervision from H.Y.H. The 4-octahedra framework was developed and implemented by E.A.M., with input from B.H.G. Structural relaxations and calculations by DFT were performed by E.A.M., Y.-F.Z. and A.S.B. Electron microscopy measurements were conducted by L.B., with input from N.S., B.H.G. and D.A.M. The manuscript was written by L.B., B.H.G., E.A.M., A.S.B. and J.A.M., with input from all authors.

**Funding** Open access funding provided by Max Planck Society.

**Competing interests** The authors declare no competing interests.

**Additional information**
**Correspondence and requests for materials** should be addressed to Lopa Bhatt or Berit H. Goodge.

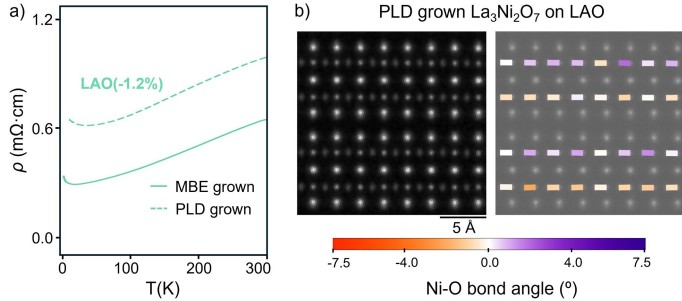

**Extended Data Fig. 1 | Electronic transport and Ni–O bond angle measurement of PLD-grown La₃Ni₂O₇ on LAO thin film. a**, Resistivity $\rho(T)$ of $La_3Ni_2O_7$ thin films epitaxially strained on LAO grown using MBE (solid line) and PLD (dashed line) as described in Methods. The transport curves show similar behaviour between the films grown using MBE and PLD. **b**, Ptychographic reconstruction (left) and corresponding Ni–O bond angle map (right) of a PLD-grown $La_3Ni_2O_7$ film on LAO. The Ni–O coordination observed in MBE-grown $La_3Ni_2O_7$ films on LAO (Fig. 2b) are similar to the structure observed in the films grown by PLD.

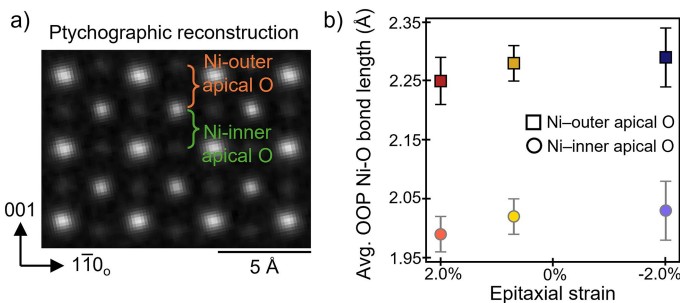

a) Ptychographic reconstruction

Ni-outer apical O

Ni-inner apical O

$1\bar{1}0_o$

5 Å

b)

**Extended Data Fig. 2 | Ni–apical O bond length change across epitaxial strain in La$_3$Ni$_2$O$_7$ thin films. a**, Experimental MEP reconstruction of La$_3$Ni$_2$O$_7$ thin film along pseudo-cubic projection. Ni–inner apical O and Ni–outer apical O bonds are highlighted. **b**, Average out-of-plane Ni–apical O bond lengths measured as a function of nominal epitaxial strain through MEP.

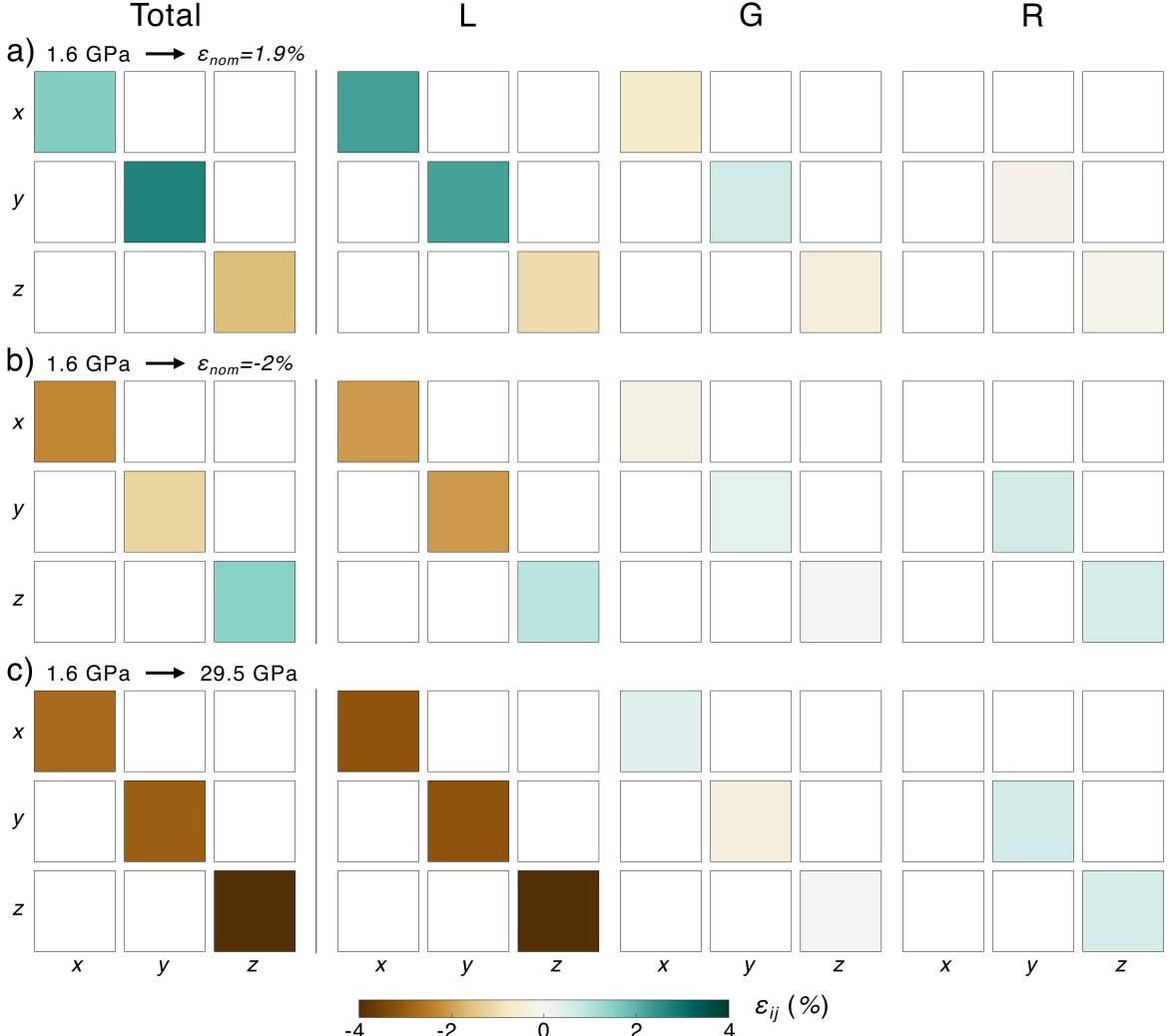

**Extended Data Fig. 3 | Partial strain tensors corresponding to $L$, $G$ and $R$ octahedral distortions. a–c,** Strain tensors are decomposed from the initial ($I$) 1.6-GPa structure reported in ref. 1 into three final structures: $\varepsilon_{nom} = 1.9\%$ obtained from relaxing $I$ with the in-plane lattice parameters of STO (**a**); $\varepsilon_{nom} = -2.0\%$ obtained by relaxing the 29.5-GPa high-pressure structure from ref. 1 with the in-plane lattice parameters of SLAO (**b**); and the 29.5-GPa high-pressure structure (**c**). In each case, the first column shows the total strain tensor for the process, whereas the $L$, $G$ and $R$ columns show the tensors owing to octahedral bond length changes, internal angular changes and rigid rotations, respectively. Higher-order mixed terms are not shown. The component matrices do not individually correspond to experimentally measured values; comparisons can be derived from the complete octahedral parameters in Supplementary Fig. 19.

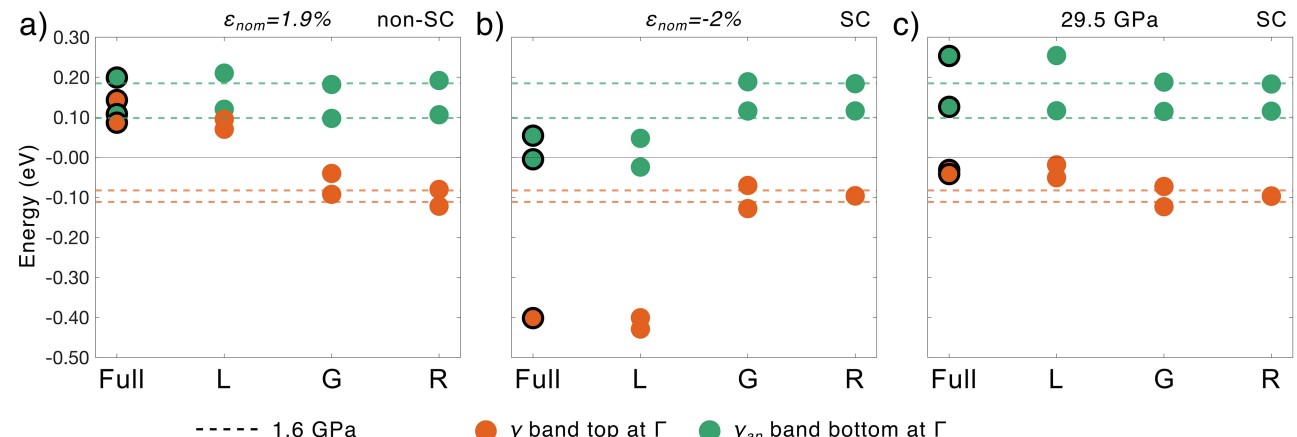

**Extended Data Fig. 4 | Low-energy band evolution for octahedral distortion-decomposed virtual structures.** Energy positions of the bonding $\gamma$ band top (orange) and the anti-bonding $\gamma_{an}$ band bottom (green) at the $\Gamma$ point extracted from the electronic structure calculations of La$_3$Ni$_2$O$_7$ with $U = 4$ eV on the Ni site, the same calculations presented in Fig. 4. The band positions for the structures derived from octahedral bond length changes (L) closely match those of the full structures. By contrast, the structures associated with internal angle distortions (G) and rotational changes (R) exhibit an energy landscape more similar to that of the reference 1.6-GPa configuration.

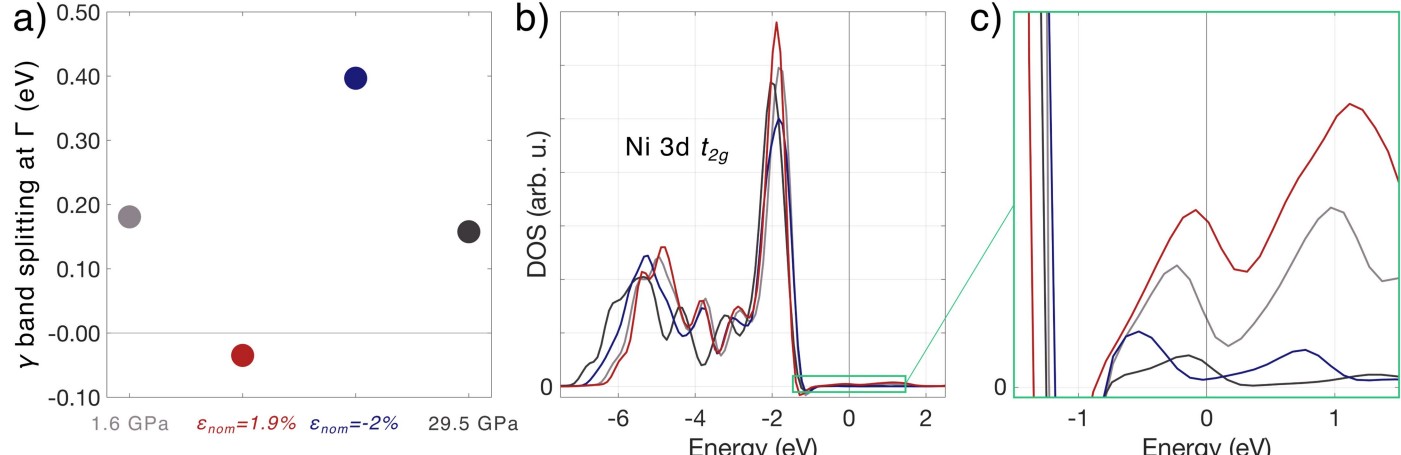

**Extended Data Fig. 5 | Band splitting and densities of states for different structural conditions. a**, $\gamma$ band splitting at $\Gamma$ extracted from the band structures in Fig. 4. The splitting is defined as the smallest energy separation between nominally hole-like and electron-like bands. The negative splitting for $\varepsilon_{nom}$ = +1.9% indicates a band crossing. **b**, Densities of states projected to the global $t_{2g}$ manifold of Ni $3d$ states. **c**, Detail of the densities of states around the Fermi level, showing the reduction (increase) of the residual $t_{2g}$ orbital content in the (non-)superconducting structures.

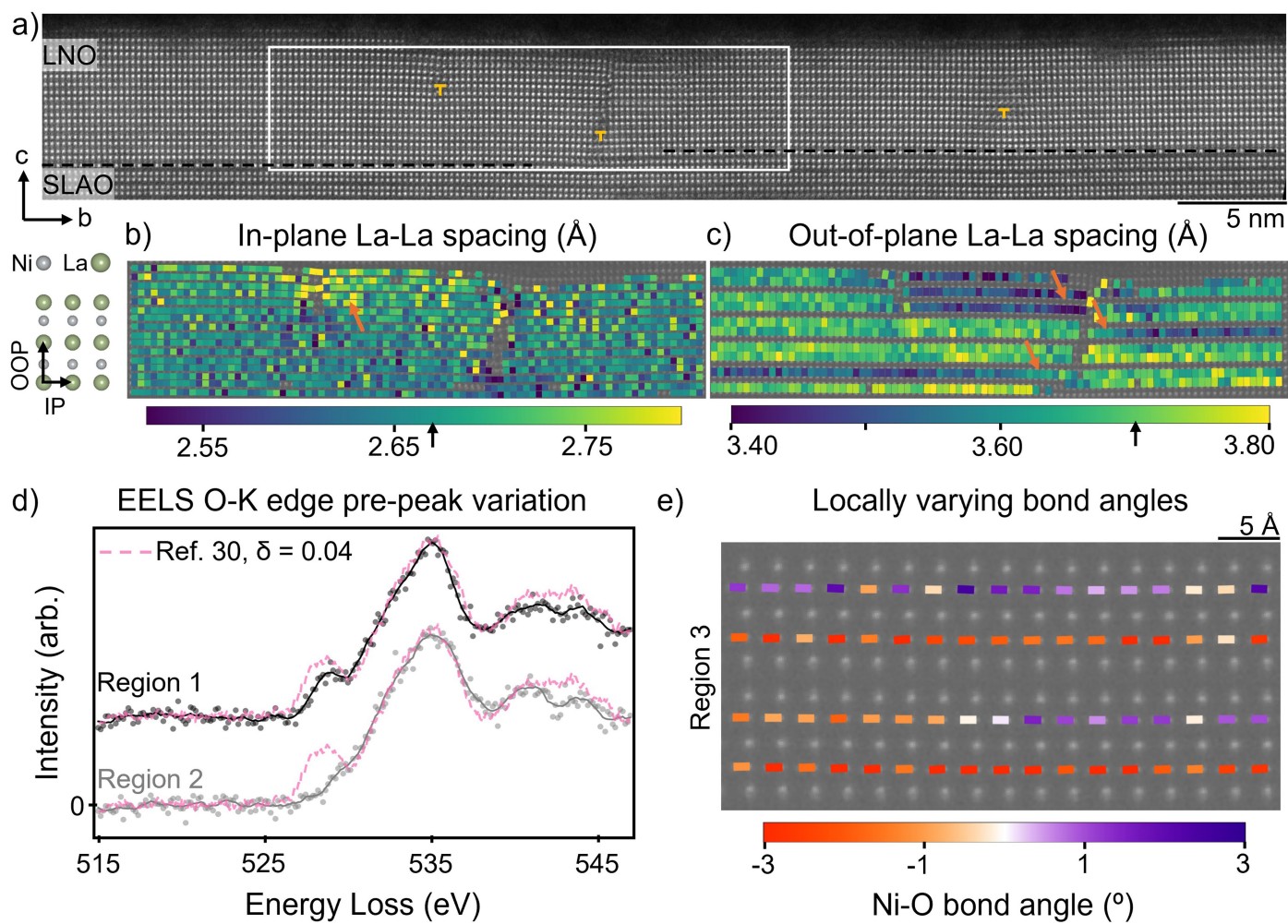

**Extended Data Fig. 6 | Impact of local inhomogeneities on lattice and oxygen structure. a**, Large-field-of-view ADF-STEM image of the $La_3Ni_2O_7$ film on SLAO. Dashed lines mark the interface between film and substrate. Yellow Ts mark crystalline dislocations. **b,c**, In-plane (**b**) and out-of-plane (**c**) La–La spacing measured by quantitative atom tracking in the region marked by the white box in **a**. Orange arrows mark relaxation near defects. Average in-plane and out-of-plane La–La spacings in clean regions of the film are marked by black arrows on the colour bar. **d,e**, O K-edge electron energy loss near edge structure (**d**) and Ni–O bond angle map (**e**) from three different regions of the same $La_3Ni_2O_7$ film on SLAO; ADF images of regions 1 and 2 are shown in Supplementary Fig. 7. Dotted pink line in **d** compares the O K-edge measured in bulk $La_3Ni_2O_{7-\delta}$ with $\delta \approx 0.04$ from ref. 10.

**Extended Data Table 1 | Tight-binding parameters (eV) of bilayer La₃Ni₂O₇ for different conditions**

| System | $\Delta E$ | $t_\perp^z$ | $t_\parallel^x$ | $t_\parallel^{xz}$ | $t_\parallel^z$ | $t_\perp^z/t_\parallel^x$ |
|---|---|---|---|---|---|---|
| 1.6 GPa | 0.366 | -0.545 | -0.401 | -0.191 | -0.075 | 1.359 |
| $\varepsilon_{\mathrm{nom}} = +1.9\%$ | 0.209 | -0.531 | -0.390 | -0.211 | -0.112 | 1.362 |
| $\varepsilon_{\mathrm{nom}} = -2.0\%$ | 0.632 | -0.591 | -0.474 | -0.204 | -0.079 | 1.247 |
| 29.5 GPa | 0.390 | -0.636 | -0.493 | -0.241 | -0.116 | 1.290 |

$\Delta E$ denotes the on-site energy between $d_{x^2-y^2}$ and $d_{z^2}$ orbitals. The superscript $z$ ($x$) indicates hopping between $d_{z^2}$ ($d_{x^2-y^2}$) and $d_{z^2}$ ($d_{x^2-y^2}$) orbitals and $xz$ represents the hopping between $d_{x^2-y^2}$ and $d_{z^2}$ orbitals. The subscript $\perp$ and $\parallel$ represent interlayer and intralayer hoppings. The 1.6-GPa and 29.5-GPa structures are obtained by relaxing refinements given in ref. 1.

**Extended Data Table 2 | Tight-binding parameters (meV) of bilayer La$_3$Ni$_2$O$_7$ for different conditions**

| System | $t_\parallel^{xy-(x^2-y^2)}$ | $t_\perp^{xy-(x^2-y^2)}$ | $t_\parallel^{xy-z^2}$ | $t_\perp^{xy-z^2}$ | $t_\parallel^{xz-yz}$ | $t_\perp^{xz-yz}$ |
|---|---|---|---|---|---|---|
| 1.6 GPa | -6.872 | 0.000 | -0.748 | 6.673 | 86.352 | 0.000 |
| $\varepsilon_{nom} = +1.9\%$ | -17.827 | 0.000 | -2.521 | 7.863 | 58.064 | 0.000 |
| $\varepsilon_{nom} = -2.0\%$ | -3.653 | 0.000 | -0.540 | 2.890 | 106.937 | 0.000 |
| 29.5 GPa | -3.501 | 0.000 | -0.315 | 1.043 | 94.032 | 0.000 |

The superscript $xy - (x^2 - y^2)$ represents the hopping between $d_{xy}$ and $d_{x^2-y^2}$ orbitals; $xy - z^2$ represents the hopping between $d_{xy}$ and $dz^2$ orbitals; $xz - yz$ represents the hopping between $d_{xz}$ and $d_{yz}$ orbitals. The subscript $\perp$ and $\parallel$ represent out-of-plane and in-plane hoppings. The 1.6-GPa and 29.5-GPa structures are obtained by relaxing refinements given in ref.1.