## [Peer Review file · Nature]

Structural modifications in strain-engineered bilayer nickelate thin films

Corresponding Author: Dr Berit Goodge

Version 0:

Reviewer comments:

Referee #1

(Remarks to the Author)

This manuscript investigates epitaxially strained $\text{La}_3\text{Ni}_2\text{O}_7$ (327) thin films and reveals specific atomic structure motifs that enable ambient-pressure superconductivity (SC). By employing multislice electron ptychography (MEP), the authors identify a structural transition from A_{2mm} symmetry in tensile-strained 327 to $Fm\bar{3}m$ symmetry in compressively strained 327. They highlight two critical factors in achieving SC in 327 films: the $Fm\bar{3}m$ symmetry and in-plane compression. Additionally, they observe local inhomogeneity and strain relaxation, which may contribute to the broad SC transition in 327 thin films.

Overall, the manuscript provides a comprehensive analysis of how epitaxial strain influences the atomic structure of 327.

The experimental results appear robust and compelling. However, the study focuses solely on the structural aspects of the SC phase in 327 thin films, which is arguably an oversimplification. For example, the authors identify the same $Fm\bar{3}m$ symmetry in 327 films subjected to -2% and -1.2% strain, yet SC is observed only in the former. Given that the 327 thin films maintain the same lattice symmetry, a mere 0.8% strain difference should not result in such a dramatic change in SC properties (details of this point can be found in Comment 2). The classical cuprate, $(\text{LaSr})_2\text{CuO}_4$, exhibits a robust SC state under both tensile strain (on STO substrates) and compressive strain (on SLAO substrates), with only a modest T_c enhancement of ~10 K under compressive strain (e.g., Phys. Rev. Lett. 89, 107001 (2002)). Is the 327 nickelate film different from cuprates? Or perhaps factors other than strain play more critical roles in forming SC states in 327 films?

By attributing the SC solely to the $Fm\bar{3}m$ structure or compressive strain (as implied by the title), the authors overlook other crucial factors, such as the roles of the SLAO substrate, interface-induced charge transfer or carrier doping, and electron-phonon coupling at the interface. These factors have been widely discussed in interface-enhanced superconducting systems and could play a significant role here as well. Although this system exhibit SC in bulk crystals under hydrostatic pressure and comparative microstructural studies of the films and bulk crystals can yield critical insights into the origin of SC, the link between the SC and atomic structure revealed in this manuscript is still too weak. Therefore, I argue that this paper serves better as a study of "the structural basis" for SC in 327 thin films rather than establishing "the structural origin" of SC. Additionally, some of the arguments in this manuscript warrant further scrutiny (shown below). Given these concerns, this work may be considered for a more specialized journal rather than Nature.

Below are specific comments:

Major Comments

1. The authors argue that in-plane (IP) compression plays a more significant role than out-of-plane (OOP) compression, primarily by analyzing the evolution of the IP and OOP lattice constants. While I agree that the IP lattice constant likely correlates with IP bonding strength, the OOP lattice constant c does not directly reflect OOP bonding strength. Instead, the authors should examine the evolution of the OOP Ni-O-Ni bond length. This aspect is particularly well-suited for investigation via STEM imaging, as XRD refinements may not be feasible for thin films. Furthermore, the relative evolution of IP and OOP Ni-O-Ni bond length is crucial in modulating the e_g orbital energy splitting, which can directly impact SC properties (see Nat Commun 16, 1054 (2025), discussion points 4 & 5). I suggest that the authors discuss this relationship in more details.

2. Fig. 3: The comparison between IP and OOP compression relative to bulk 327 is problematic. The dashed line in Fig. 3 more accurately represents a critical lattice parameter associated with the symmetry transition into the $Fm\bar{3}m$ phase, within which superconductivity is realized. However, the authors appear to interpret it as a 'critical IP lattice constant for SC', without fully considering the symmetry change.

Notably in Fig. S8 of ref. 4, La₂PrNi₂O₇, which has a smaller IP lattice constant (by 1%-2%), requires a higher critical pressure to become superconducting. This observation contradicts the authors' hypothesis. To date, SC in pressurized RP nickelates has consistently emerged in conjunction with a symmetry transition.

Given this, the role of IP compression may be primarily to stabilize the high-symmetry structure, rather than being directly responsible for SC—unless the authors can provide compelling evidence that the IP lattice constant influences SC within the Fmmm/Immm phase. For example, I would expect the IP lattice constant of superconducting La₂PrNi₂O₇ thin films (arXiv:2501.08022) to be significantly larger than that of bulk La₂PrNi₂O₇ at P_c (5.30 Å in Ref. 4, compared to 5.34 Å in parent 327), which would contradict the authors' expectations. Additionally, in bulk 327, once the Fmmm/Immm phase is reached, further pressure increases lead to a suppression of T_c (see Ref. 35), which is also inconsistent with the authors' claims. If this comparison is not well supported, I suggest removing it.

3. The authors frequently describe 327 as 'cuprate-like', implying a dx₂-y₂-driven SC mechanism based on the absence of the γ band (primarily the bonding dz₂ band) in compressively strained films (as calculated in Refs. 27-28). However, even under this assumption, the Fermi surface topology of 327 remains fundamentally different from that of cuprates, as it still involves both dx₂-y₂ and dz₂ components.

Rather than claiming cuprate-like behavior, the authors can more precisely argue that SC in thin films is unlikely to be driven by the metallization of bonding dz₂ orbitals. This is already a strong constraint on theoretical models. To support their discussion, I suggest that the authors include DFT calculation of the electronic structure based on their measured or relaxed Fmmm structure, highlighting the Fermi surface topology. They should also address the implications of the absence of a metallized γ band (i.e. absence of a hole-doped dz₂ orbital). Useful references include Phys. Rev. B 111, 014512 (2025) and the sources cited therein.

Additionally, a recent ARPES study on SC 327 thin films unexpectedly reveals a metallized γ band (arXiv:2501.09255), which is confusing. The authors may comment on the discrepancy between ARPES-measured band structures and their DFT-calculated results.

4. The authors should ensure that they are not inadvertently measuring a twinned domain along the b-axis in the compressed films, as suggested in the tensile strained films by the blue region in Fig. S14. One way to verify this is by comparing the [010]-projected MEP image of tensile-strained films with Fig. 2a-b and discussing the differences. This concern arises because a slight vertical elongation of oxygen atoms in Fig. 2a-b is noticed, compared to Fig. 2c-d.

5. The MEP results are impressive. Could the authors provide details on the optimized parameters from Bayesian optimization used in their final reconstructions (e.g. regularization, any real-space super-sampling, point spread function, etc.) ?

6. In Fig. 2f, for SLAO and LAO, please consider separating the bond angles from the upper and lower layers. Presenting these as two distinct components within a histogram using different colors would improve clarity, as the current two-Gaussian fitting is not so well-defined.

7. In Fig. 4a-c, it appears that all T markers indicate stacking faults associated with transitions between RP phases with n=1 and n=2. Do the authors imply that strain relaxation occurs through the intergrowth of n=1 RP phases? If so, please clarify this at the beginning of the section to avoid potential misunderstandings. Such defects may be detrimental instead of beneficial to SC since larger strain may not lead to better SC. Some of the statements in page 13 should be adjusted.

8. In recent preprints, such as arXiv: 2412.16622, resistivity of similar 327 films can change significantly during the measurements especially when the temperature is high, which was attributed to the oxygen loss. Did the authors see similar effects? For TEM measurements, is it possible to see any hints of oxygen loss during the sample preparation or sample degradation? What is the origin of the inhomogeneous oxygen stoichiometry shown in Fig. 4d? Is the O-K statistically preferred as that from the region 1 or region 2?

Minor Comments

9. Page 7: "a focused probe" may be changed to "an overfocused probe" or "a convergent probe" to avoid ambiguity.

10. Fig. 1c: why are the R-T curves for films on LAO and NGO only shown down to 50 K? This temperature is even higher than the onset T_c in SLAO.

11. The manuscript refers to bond angles, but these appear to be measured as O-Ni-Ni rather than Ni-O-Ni. If so, they may be more appropriately described as "octahedral tilt angles". Please consider revising the terminology for clarity.

12. Page 10: the phrase "in-plane lattice constant and octahedral symmetry as order parameters for superconductivity" is misleading. In the context of Ginzburg-Landau theory, the order parameter of a superconductor has a specific definition. Consider rewording this as "determining factors for superconductivity" instead.

Referee #2

(Remarks to the Author)

The authors describe a systematic structural study using a combination of STEM, MEP and XRD as a function of epitaxial strain after thin film growth on substrates with varying in-plane lattice parameters. They observe two trends as the in-plane lattice parameter approaches a -2% strain, where superconductivity is observed. The first trend is an increase of the planar Ni-O bond angles, while the 2nd weaker trend is an increase in the out-of-plane lattice parameter. These structural trends are used to conclude a "...necessity of in-plane rather than c-axis compression for achieving superconductivity." The structural measurements are excellent and well-presented. Before recommending for publication, there are a number of weaknesses that need to be addressed.

1) The most obvious correlation between the data is that PLD grown films are superconducting and MBE grown films are not. Please include transport and structural data on MBE-grown films on SLAO.

2) The trends in the transport could be discussed. There doesn't seem to be an obvious correlation between bond angle and conductivity.

3) What do measurements of carrier density based on magneto transport show?

- 4) The authors point out that the microscopy shows inhomogeneities (Fig. 4), which may lead to filamentary superconductivity in these PLD grown films. But where is the superconductivity taking place? What is needed is a discussion of the possibility that the areas with smaller out-of-plane La-La spacing are superconducting. A figure similar to Fig. 4 for the thin films grown on the other substrates would shed light on this issue.
- 5) How much of the film volume is relaxed and has a smaller out-of-plane lattice parameter? From the original paper reporting superconductivity, a nice XRD reciprocal space map is shown with a large volume of relaxed material diffracting with a smaller lattice parameter. (see extended data Figure 8 c, d and e in Ko, E.K., Yu, Y., Liu, Y. et al. Signatures of ambient pressure superconductivity in thin film La₃Ni₂O₇. Nature (2024). <https://doi.org/10.1038/s41586-024-08525-3>)
- 6) The XRD RSM would be better plotted in a way that shows a wider dynamic range of diffracted intensity to gauge relative volume of material with various lattice parameters. The weaker diffuse diffraction in Ko et al. may imply a larger volume of relaxed material than that which is coherently strained.
- 7) Some key observations reported here have been previously published, weakening the author's case of novelty. A flattening bond angle was reported in Ref. 2 of the manuscript. Superconductivity of compressively strained thin films is reported in Ref. 6 of the manuscript. The authors' appeal to novelty is based on direct real space imaging "...for the first time-" (page 4). Discussion of points 4 and 5 would build on the novel aspects of the work.

Referee #3

(Remarks to the Author)

The manuscript Resolving Structural Origins for Superconductivity in Strain-Engineered La₃Ni₂O₇ Thin Films by Bhatt et al. investigates the structure of bilayer nickelate thin films, specifically on the SLAO substrate, using various experimental techniques, including MEP/STEM, XRD, transport, EELS, and ELNES. The study aims to identify structural origins of the recently observed superconducting phase by exploring different substrates spanning from compressive to tensile strain. In contrast to earlier work, the authors suggest that in-plane compression may play a more significant role than vertical compression, potentially indicating a more cuprate-like physics in bilayer nickelates than so far assumed.

The topic addressed in this manuscript is both timely and highly relevant. Given that some of the authors were involved in the discovery of superconductivity in strained bilayer nickelates, I assume the samples analyzed here are either the same as or prepared in a comparable manner to those in the original discovery paper.

However, I have concerns that the line of argument and the conclusions drawn may not be sufficiently compelling to justify publication in Nature. Additionally, while the insights presented are certainly important, of high quality, and complementary to previous studies, they may not be as groundbreaking as those in the original discovery paper.

Below, I outline several important points that need to be addressed before further consideration of the manuscript:

- Could the authors provide a hypothesis as to why, among all the studied systems, superconductivity is observed exclusively in LNO/SLAO?
- Figure 1c shows a non-monotonic evolution of resistivity as a function of strain. In particular, the STO substrate results in insulating behavior, which contrasts with recent theoretical predictions by different groups. Could the authors comment on this discrepancy? Additionally, what structural phase is observed in films grown on STO, given that tensile strain has previously been shown to destabilize nickelate films?
- The nickelate film on SLAO is grown using a different methodology compared to the other three systems, which do not exhibit superconductivity. Could the authors justify this choice and discuss in the main text whether it could influence the observed differences?
- The authors propose that bilayer nickelates in general may exhibit physics that is more cuprate-like rather than being impacted by Ni dz² states. However, without contributions from these states, the Fermi surface remains nickelate-like, and various nickelate systems have not exhibited superconductivity for decades. Could the authors elaborate on what their proposed mechanism is, how it aligns with these considerations, and how it follows from their findings?
- A significant portion of the manuscript discusses details of the oxygen relaxation under compressive strain, particularly alternating octahedral tilts versus uniform inward/outward relaxation patterns. The authors claim to observe the latter and draw parallels to high-pressure experiments. However, the energy difference between these configurations appears to be very small, based on the DFT results in the SI, which should be presented more prominently. Could the authors clarify the impact of this structural difference on the electronic properties and whether it is substantial enough to support their claims regarding its influence on superconducting pairing? Given that strain is likely the dominant effect, a clearer discussion on this aspect would be valuable to the reader.
- The DFT calculations were performed using GGA without additional corrections, such as Hubbard terms. I suggest including DFT+U results and discussing Fig S9 more prominently in the main text to assess the robustness of the conclusions.
- Could the authors clarify the distinction between Fmmm and I4/mmm for the present a=b case, particularly in relation to the phrase "lifting of crystalline symmetry"? It seems that the symmetry in the films is actually higher than that of the ambient-pressure bulk phase. Additionally, the structures shown in Fig S7 both appear tetragonal. As the authors state, temperature

may also play an important role here. At what T have the structural measurements been performed?

- I assume that the colors in Figure 2a represent the projection of horizontal Ni-O bond angles onto a plane perpendicular to the viewing axis, rather than being related to rotations about the vertical c-axis. Could the authors confirm this interpretation and provide additional details on the representativeness of the images shown, particularly since the authors mention later that defects and interface roughness strongly impact these structural aspects.

- To improve clarity, reference spectra should be added to Figure 4d to better support the interpretation of the prepeak and its relation to oxygen concentration.

Version 2:

Reviewer comments:

Referee #1

(Remarks to the Author)

The authors have done an exceptional job in revising the manuscript, thoroughly addressing all previous comments and significantly strengthening the work with new DFT calculations. The octahedral distortion-decomposed DFT analysis is a particularly innovative and impactful addition, providing profound insights into the specific structural degrees of freedom that govern superconductivity. This approach convincingly underscores the critical importance of precise local structural measurements, elevating the interest and relevance of this work to a broad scientific audience. I am pleased to recommend this excellent manuscript for publication in Nature.

I have a few minor comments and suggestions that could further enhance the clarity and impact of the manuscript:

1. The new DFT calculations exploring the independent effects of bond length (L) and octahedral distortions (R, G) are highly valuable. However, the description of the chosen parameters could be clearer.
 - 1). Please explicitly state what the length L represents (e.g., the OOP or IP Ni-O bond length) and specify the magnitude of change applied in the calculations (e.g., $\pm 2\%$, $\pm 4\%$). If Extended Data Fig. E3 reads like that L changes up to -4% (~ 8 pm), then it is much larger than the measured value.
 - 2). It would be beneficial to discuss how these computationally tested values compare to the experimentally measured changes. For instance, the experimental OOP Ni-O bond length change between tensile and compressive strains is relatively small (~ 2 pm), whereas a -4% change in L represents a much larger distortion (~ 8 pm). This OOP bond length change is much smaller than c-axis change (40 pm, 2.5%). This may explain why the expansion along c-axis is not very important since the Ni-O bond length does not change obviously (only 2 pm).
 - 3). Similarly, a brief note on whether the magnitudes of the R and G distortions used in the DFT models are consistent with the experimentally observed ranges would be helpful.
2. The statistical analyses of O-Ni-Ni angles in Fig. 2f could be further improved. The two distinct oxygen sites within the NiO₂ plane (i.e., even and odd sites) can be treated as separate sub-lattices, since the neighboring two angles along the plane are not always equal and measured separately. This could further statistically verify the unidirectional alignment and the symmetry change in the compressive films.
3. Supplemental Table S4 effectively outlines the parameter ranges for the Bayesian optimization. For completeness and transparency, it would be useful to also state the final, optimized values for key parameters (such as convergence angle and scan step) that resulted from this process. A note on whether these values converged consistently across different datasets would further demonstrate the robustness of the methodology.
4. The reference in the caption of Fig. S11 is not correctly displayed.

Referee #2

(Remarks to the Author)

The authors have improved the manuscript with new experiments, but there are still critical gaps. In particular, there are some issues that need to be addressed.

The first is less than ideal experimental design, and it seems like the MEP imaging, according to the authors, would best be done on carefully prepared La₂PrNi₂O₇ as a function of annealing and strain.

As pointed out in initial comments, MBE films grown for this work are not superconducting while PLD films are. Referencing other groups' work is not sufficient, as specific growth and processing conditions used in those works may lead to differing strain states and superconductivity. Also, do the Pr based MBE-grown films exhibit the same progression of structure observed using multislice electron ptychography (MEP)? Perhaps there may be two essential ingredients for realizing superconductivity in the RP nickelates, or just one not captured by the present MEP study. A more optimal experimental

design would be to investigate the structure using MEP and XRD of the La₂PrNi₂O₇s as a function of the ozone annealing done by Y. Liu et al, Nature Materials volume 24, pages1221–1227 (2025). (Ref 11 in the MS).

An improved XRD analysis of the fraction of film coherently strained as well as synchrotron measurements of octahedral rotations would strengthen the conclusions. Contrary to the author's assertion that such measurements are not possible, they are now done at a number of US and international beamlines (cf, X-Ray Scattering Techniques for Epitaxial Oxide Thin Films, in "Measuring Octahedral Rotations in Perovskite Oxide Heterostructures" by S. J. May, Springer 2025). As the authors point out in the rebuttal, the use of synchrotron diffraction would be superior to lab-based diffraction. The salience of this point is indicated by the improved presentation of XRD for 327 La film on LaAlO₃ where a large volume of diffuse intensity is observed, leading one to surmise that a major fraction of the film is not coherently strained to the substrate. If so, are the MEP structural parameters intrinsic to the RP nickelates or an exceptional property of the film? The authors appear to assume a particular conclusion by stating that "Here, we have measured the precise values of O-Ni-Ni angles, bond lengths and lattice constants to isolate the structural motifs partaking in SC and provide a correct structural starting point for ab initio calculations." Electron microscopy does not identify the SC regions.

The inclusion of Hall measurements is incomplete. In the spirit of the experimental approach, measurements are best presented as a function of strain to eliminate strain induced doping effects as the dominant change with degree of strain.

The addition of a new section on first principles theory is helpful, but raises a technical question. The authors need to justify in more depth for this particular nickelate system the validity of using first principles theory to predict the electronic structure via the approach of artificially moving the atoms away from a ground state structure, the "virtual structures" on page 12.

In its current form, publication in Nature is not recommended.

Version 3:

Reviewer comments:

Referee #1

(Remarks to the Author)

The authors have adequately addressed the concerns, and the manuscript is now acceptable. I have one additional technical question regarding Supplemental Table S5: the convergence angle and scan step size differ across the listed datasets. The difference is as large as $\pm 1\%$ for convergence angle, which is significant for MEP. Could the authors clarify whether this variation stems from optimization errors or from intentional adjustments to the imaging conditions for each dataset?

Referee #2

(Remarks to the Author)

A critical weakness in the manuscript remains, which is the lack of superconductivity in MBE films grown by the team at any strain state. They need to perform the reference check that superconductivity in their MBE films occurs, in an overlapping strain range with superconducting PLD films, or the natural Occam's razor conclusion is that their MBE films are not superconducting while the PLD films are. A delay in substrate delivery is not uncommon in the field, and for an impactful study is worth the wait. Once the baseline demonstration is made, the manuscript is in much stronger position for publication.

Beyond this key issue, the authors have made meaningful improvements to the manuscript in response to earlier comments, and the study addresses a timely problem in the field of strained nickelate thin films and superconductivity. Some aspects of the structural analysis would benefit from further clarification to bolster the overall conclusions.

One point that merits additional discussion is the fraction of relaxed versus coherently strained film. While one may not expect a precise quantitative estimate, a reasonable order-of-magnitude assessment of the likely relaxed fraction would strengthen the interpretation. This is particularly relevant because the percolation threshold through optimally doped regions required for superconductivity could be small; providing a plausible range would help readers better evaluate the physical picture.

Regarding the interpretation of the X-ray diffraction (XRD) data mentioned in the rebuttal, the authors could refine their language and analysis. XRD does not simply "average" atomic positions in a volumetric sense; rather, scattering amplitudes from distinct regions combine coherently or incoherently, both of which deliver intensities characteristic of phases present in the sample (such as the superlattice peaks seen in Fig. 2e). Clarifying this point—and its implications for strain heterogeneity—would improve the rigor of the structural argument.

The question of whether the coherently strained fraction of the film is intrinsically superconducting is central to the manuscript. Relevant recent ARPES work (arXiv:2502.17831) the authors could include in a revised manuscript suggests a superconducting gap opening below T_c . At the same time, other recent studies (e.g., arXiv:2504.16372v2) highlight open questions about the electronic structure and the nature of the gap above T_c . Framing the manuscript in this broader and still-evolving context—acknowledging both supportive and critical evidence—would make the claims more balanced and

persuasive.

The authors are commended for including the Hall data, which adds relevant information about charge transport and doping. This is a welcome addition. However, the manuscript's use of these data to argue for close similarity between PLD- and MBE-grown films is an overly strong claim. Given known differences between these growth techniques, a more nuanced discussion of similarities and potential differences would be appropriate.

Once superconductivity is demonstrated in the MBE-grown films, this becomes a promising contribution to the field. Additionally, with targeted revisions that clarify the strained versus relaxed fractions, refine the XRD interpretation, and more carefully situate the superconductivity claim within the current literature, the manuscript has the potential to make a meaningful impact.

We thank all three Referees for their effort and time to review our work, and for their comments and suggestions to improve it. Below we include point-by-point responses; referee comments are in **black**, our responses are in **blue**, and changes to the text are in **green**.

Referee #1 (Remarks to the Author):

This manuscript investigates epitaxially strained $\text{La}_3\text{Ni}_2\text{O}_7$ (327) thin films and reveals specific atomic structure motifs that enable ambient-pressure superconductivity (SC). By employing multislice electron ptychography (MEP), the authors identify a structural transition from A_{2mm} symmetry in tensile-strained 327 to $Fm\bar{3}m$ symmetry in compressively strained 327. They highlight two critical factors in achieving SC in 327 films: the $Fm\bar{3}m$ symmetry and in-plane compression.

Additionally, they observe local inhomogeneity and strain relaxation, which may contribute to the broad SC transition in 327 thin films. Overall, the manuscript provides a comprehensive analysis of how epitaxial strain influences the atomic structure of 327. **The experimental results appear robust and compelling.**

We thank the reviewer for their positive assessment of our work and its relevance to the community.

However, the study focuses solely on the structural aspects of the SC phase in 327 thin films, which is arguably an oversimplification. For example, the authors identify the same $Fm\bar{3}m$ symmetry in 327 films subjected to -2% and -1.2% strain, yet SC is observed only in the former. Given that the 327 thin films maintain the same lattice symmetry, a mere 0.8% strain difference should not result in such a dramatic change in SC properties (details of this point can be found in Comment 2).

We thank the reviewer for raising this point. The effect of such small strain on the electronic properties is indeed very surprising and the question of “critical strain” for superconductivity (SC) in $\text{La}_3\text{Ni}_2\text{O}_7$ is so far unresolved. However, we would like to emphasise that a symmetry comparison alone between films on LAO and SLAO is not sufficient to fully capture the differences between the two systems: the magnitudes of lattice spacing and Ni-O plane buckling are distinct between the two films and should be taken into account.

In Figure R1, we reproduce a comparison of reported $T_{c,\text{onset}}$ vs in-plane (IP) lattice constant of thin films on SLAO and bulk samples from across the literature (to appear in Y. Liu et al. Nature Materials, *in press*). This suggests that the nominal IP lattice constants for films fully strained to LAO substrates ($a, b = 3.787 \text{ \AA}$ with no relaxation) may be enough to stabilize SC. But, as discussed in Fig. 3 of the main text, the experimentally measured lattice constants for our films on LAO (both MBE and PLD-grown) currently fall just outside of this range due to the minor strain relaxation and defect formation which is discussed in Extended Data Fig. E6. Further improvement of the film quality and exploration to larger

compressive strains will help address this point raised by the reviewer, but falls beyond the scope of this work.

[REDACTED]

We have added the following comment:

“A survey of the in-plane lattice spacings versus T_c for $\text{La}_3\text{Ni}_2\text{O}_7$ films grown on SLAO shows an absence of superconducting transitions for films with IP lattice constant larger than $\sim 3.79 \text{ \AA}$ [E. K. Ko et al. *Nature* 630 (2025), Y. Liu et al. *Nature Materials* - *in press*], very near the LAO lattice spacing of 3.787 \AA . This suggests that further improvement in sample quality on LAO may be able to stabilize superconductivity in films with no epitaxial relaxation and low concentration of defects, but the precise critical strain for superconductivity in $\text{La}_3\text{Ni}_2\text{O}_7$ remains to be robustly established, as does the effect of even higher compressive strains.”

The classical cuprate, $(\text{LaSr})_2\text{CuO}_4$, exhibits a robust SC state under both tensile strain (on STO substrates) and compressive strain (on SLAO substrates), with only a modest T_c enhancement of $\sim 10 \text{ K}$ under compressive strain (e.g., *Phys. Rev. Lett.* 89, 107001

(2002)). Is the 327 nickelate film different from cuprates? Or perhaps factors other than strain play more critical roles in forming SC states in 327 films?

Indeed, this question is one at the heart of current research in SC nickelates. Our structural analysis and DFT calculations suggest that not only the change in the lattice parameters but also Ni-O bond coordination play a role in considerably changing the electronic band structure. While we are as yet unable to answer the overarching question if nickelates are the same as cuprates, DFT calculations presented in the paper indicate that structural factors other than changes in lattice constant alone may play critical roles in stabilizing SC in $\text{La}_3\text{Ni}_2\text{O}_7$ films.

We also wish to highlight additional oxide superconductors which are sensitive to strain or pressure, such as infinite-layer nickelates [N. N. Wang et al. *Nat Commun* 13 (2022), Y. Lee et al. *Nature Synthesis* 4 (2025)], Sr_2RuO_4 [H. Nair et al. *APL Mater* 6 (2012), C. W. Hicks et al. *Science* 344 (2014)], and RuO_2 [J. P. Ruf et al. *Nat Commun* 12 (2021)], though we note that the mechanism of strain enhancement varies between systems – for example, in Sr_2RuO_4 the large fractional enhancement of T_c under uniaxial pressure is due to the driving of a Lifshitz transition, while hydrostatic pressure on the same compound has relatively little effect on T_c . The reference mentioned by the Reviewer (*Phys. Rev. Lett.* 89, 107001 (2002)) also highlights the huge experimental challenge of robustly disentangling various parameters such as epitaxial strain and oxygen stoichiometry which are linked through synthesis. With a combination of high-precision local structural measurements and our DFT structural decomposition approach, we are able to address some of these challenges in $\text{La}_3\text{Ni}_2\text{O}_7$.

By attributing the SC solely to the Fmmm structure or compressive strain (as implied by the title), the authors overlook other crucial factors, such as the roles of the SLAO substrate, interface-induced charge transfer or carrier doping, and electron-phonon coupling at the interface. These factors have been widely discussed in interface-enhanced superconducting systems and could play a significant role here as well.

We thank the reviewer for raising the possibilities relating to the particular choice of substrate. From our microscopic studies, we have observed a variety of different atomic structures at the interface (as described in E. K. Ko et al. *Nature* 630 (2025), Y. Liu et al. *Nature Materials* - in press (arXiv:2501.08022) and this work), also reported by other groups [G. Zhuo et al. *Nature* 640 (2025) and H. Wang et al. arXiv:2502.18068 (2025)]. This large variability in interface atomic structure among SC thin films suggests that a specific atomic structure is not alone responsible for the observed phenomena. Furthermore, recent reports of SC in thin films grown on LAO, NGO and STO under hydrostatic pressure points to the irrelevance of specific substrate-film interface needed in realizing SC in $\text{La}_3\text{Ni}_2\text{O}_7$ [M. Osada et al. *Communications Physics* 8 (2025)].

With the recent and significant improvement in sample quality achieved through Pr substitution, we have observed a large increase in critical current density J_c . Ginzburg-Landau fitting extracted superconducting thickness is in good agreement with film thicknesses extracted by Scherrer analysis of XRD and ADF-STEM imaging, pointing

towards bulk-like SC in our films [Y. Liu et al. *Nature Materials* - in press (arXiv:2501.08022)].

We have included the text below to highlight the potential importance of interface:

A variety of different interfacial atomic structures are observed by microscopic studies here and elsewhere [E. K. Ko et al. *Nature* 630 (2025), Y. Liu et al. *Nature Materials* - in press (arXiv:2501.08022), G. Zhuo et al. *Nature* 640 (2025), H. Wang et al. arXiv:2502.18068 (2025)], and thin films grown on LAO, NGO, and STO under hydrostatic pressure have also shown superconducting transitions [M. Osada et al. *Communications Physics* 8 (2025)]. Superconductivity therefore appears to be largely interface-independent, though more subtle effects remain to be explored [B. Geisler et al. arXiv:2503.10902 (2025), C. Le et al. arXiv:2501.14665 (2025)].

Although this system exhibit SC in bulk crystals under hydrostatic pressure and comparative microstructural studies of the films and bulk crystals can yield critical insights into the origin of SC, the link between the SC and atomic structure revealed in this manuscript is still too weak. Therefore, I argue that this paper serves better as a study of “the structural basis” for SC in 327 thin films rather than establishing “the structural origin” of SC.

We appreciate the reviewer’s suggestion and have updated the title of our manuscript accordingly: “Strain Engineering Bilayer Nickelates to Resolve the Structural Basis for Superconductivity”

Additionally, some of the arguments in this manuscript warrant further scrutiny (shown below). Given these concerns, this work may be considered for a more specialized journal rather than Nature.

Below are specific comments:

Major Comments

1. The authors argue that in-plane (IP) compression plays a more significant role than out-of-plane (OOP) compression, primarily by analyzing the evolution of the IP and OOP lattice constants. While I agree that the IP lattice constant likely correlates with IP bonding strength, the OOP lattice constant c does not directly reflect OOP bonding strength. Instead, the authors should examine the evolution of the OOP Ni-O-Ni bond length. This aspect is particularly well-suited for investigation via STEM imaging, as XRD refinements may not be feasible for thin films.

Furthermore, the relative evolution of IP and OOP Ni-O-Ni bond length is crucial in modulating the e_g orbital energy splitting, which can directly impact SC properties (see Nat Commun 16, 1054 (2025), discussion points 4 & 5). I suggest that the authors discuss this relationship in more details.

We appreciate the reviewer's suggestion, and now include measurements of the out-of-plane (OOP) Ni-O spacing measured from ptychography reconstructions along the $[100]_{pc}$, separated by Ni to inner apical O (circle) and Ni to outer apical O (square). A clear increasing trend in the OOP Ni-O spacing is observed, consistent with the c -axis expansion with increasing compressive strain. The ratio between the two bond lengths (Ni-outer apical O / Ni-inner apical O) remains relatively constant across the strain series.

Extended Data Fig. E2 | (a) Experimental multislice electron ptychographic reconstruction of an $\text{La}_3\text{Ni}_2\text{O}_7$ thin film along the pseudo-cubic $[100]_{pc}$ projection. Ni-inner apical O and Ni-outer apical O bonds are highlighted. (b) Average out-of-plane (OOP) Ni-apical O bond lengths measured as a function of nominal epitaxial strain via MEP separated by inner (circle) and outer (square) oxygens.

The ratio of OOP to IP lattice spacing also increases with increasing compressive strain in thin films (opposite to the observed trend in bulk crystals under hydrostatic pressure), as shown below:

Fig. R2 | Average OOP to IP lattice spacing ratio experimentally measured across the strain series.

Based on our structural measurements, we have calculated inter- and intra-layer hopping parameters from our DFT models for compressive and tensile strained films on SLAO and STO, as well as for previously published structures for the bulk phase under low- and high-pressure conditions [H. sun et al. *Nature* 621 (2023)]. These calculated parameters are now included in Extended Data Table E1, E2 and agree with previously published values [Y. Zhang et al. *Nature Communication* 15 (2024), Y. Zhao and A. S. Botana *Phys. Rev. B* 111 (2025)]. Notably, we find that the ratio of inter-layer d_{z2} hopping t_{\perp}^z to intra-layer d_{x2-y2} hopping t_{\parallel}^x which has been previously discussed in the context of pairing symmetry [C. Lu et al. *Phys. Rev. Lett.* 132 (2024), Y. Zhao and A. S. Botana *Phys. Rev. B* 111 (2025), C. Xia et al. *Nature Communication* 16 (2025)] decreases upon compressive strain in thin films, as well as upon hydrostatic pressure in bulk despite opposite evolutions of their OOP bond lengths. In contrast, the IP hybridization between the d_{x2-y2} and d_{z2} orbitals inferred from the ratio of t_{\parallel}^{xz} to t_{\parallel}^x shows opposite evolution between bulk and thin film.

We have added this discussion in the main text:

“Our results, however, clearly show that *c*-axis – and in particular OOP Ni-O bond – compression is not necessary to achieve superconductivity in $\text{La}_3\text{Ni}_2\text{O}_7$ thin films, raising questions about the relevant electronic bands which participate in superconductivity. Instead, hopping parameters calculated from our structural models of compressive and tensile strained films on SLAO and STO as well as for previously published structures for the bulk phase under low- and high-pressure conditions (Extended Data Tables E1, E2) indicate that the ratio of inter-layer d_{z2} hopping t_{\perp}^z to intra-layer d_{x2-y2} hopping t_{\parallel}^x which has been previously discussed in the context of pairing symmetry [C. Lu et al. *Phys. Rev. Lett.* 132 (2024), Y. Zhao and A. S. Botana *Phys. Rev. B* 111 (2025), C. Xia et al. *Nature Communication* 16 (2025)] decreases both upon compressive strain in thin films as well as upon hydrostatic pressure in bulk, despite opposite evolutions of their OOP bond lengths.”

2. Fig. 3: The comparison between IP and OOP compression relative to bulk 327 is problematic. The dashed line in Fig. 3 more accurately represents a critical lattice parameter associated with the symmetry transition into the Fmmm phase, within which superconductivity is realized. However, the authors appear to interpret it as a ‘critical IP lattice constant for SC’, without fully considering the symmetry change.

We appreciate the opportunity to clarify this point: the reviewer is correct that the relevance of the dashed line should be related to the observed change in crystal symmetry concomitantly with the SC transition. We have clarified the meaning of the dashed line by:

“The IP spacing of the superconducting film on SLAO closely matches the IP spacing of bulk samples at the onset of superconductivity and structural transition (dashed line).”

Notably in Fig. S8 of ref. 4, La₂PrNi₂O₇, which has a smaller IP lattice constant (by 1%-2%), requires a higher critical pressure to become superconducting. This observation contradicts the authors' hypothesis. To date, SC in pressurized RP nickelates has consistently emerged in conjunction with a symmetry transition. Given this, the role of IP compression may be primarily to stabilize the high-symmetry structure, rather than being directly responsible for SC—unless the authors can provide compelling evidence that the IP lattice constant influences SC within the Fmmm/I4mmm phase.

For example, I would expect the IP lattice constant of superconducting La₂PrNi₂O₇ thin films (arXiv:2501.08022) to be significantly larger than that of bulk La₂PrNi₂O₇ at P_c (5.30 Å in Ref. 4, compared to 5.34 Å in parent 327), which would contradict the authors' expectations.

We would first like to clarify that all of the analysis presented here is for the stoichiometric compound La₃Ni₂O₇, and has not yet been extended to include other rare earth substitutions which will correspondingly change the crystalline lattice constants, as pointed out by the Reviewer. We would also like to emphasize that the reference highlighted by the Reviewer is just one of many P-T phase diagrams reported in the literature, and that precise pressure at which the SC appears varies significantly between individual reports. Many factors, including the quality of samples – especially in regards to impurity phases and oxygen stoichiometry – impact both structural and electronic measurements reported in phase diagrams throughout the literature. Hence, we caution against such direct comparison of precise pressure and lattice spacing values in bulk until further improvement in sample quality and measurement techniques can provide a more precise bulk La₃Ni₂O₇ phase diagram.

We agree with the reviewer that in bulk samples thus far, SC is observed alongside a structural transition. The structural transition boundary, however, has been reported over a large pressure range, which obfuscates concrete conclusions regarding an interplay between structural transition and SC.

In thin films, our measurements show that both compressively strained films on LAO and SLAO have similar high-symmetry structure. Thus far, however, no superconductivity has been found in as grown La₃Ni₂O₇ thin films on LAO at ambient pressure, although recent reports demonstrate that SC in thin films on LAO can be stabilised through application of hydrostatic pressure [M. Osada et al. *Communications Physics* 8 (2025)]. This indicates that *both* compressed bond lengths and modification of bond angles influence the ability to stabilize a superconducting transition within the same symmetry structure.

Experimentally, it is not possible to disentangle the contribution of lattice spacings and bond angles changes, motivating our addition of DFT strain decomposition analysis, now included as Figure 4 in the main text and related discussion (see response to #3 below). We have updated the wording in the paper at various places to avoid suggesting that in-plane lattice spacing alone can explain the influence on SC.

Additionally, in bulk 327, once the Fmmm/I4mmm phase is reached, further pressure increases lead to a suppression of T_c (see Ref. 35), which is also inconsistent with the authors' claims. If this comparison is not well supported, I suggest removing it.

We appreciate the opportunity to clarify our discussion here. Dome-like behavior is common to high- T_c SC and does not contradict the role of structural changes examined in this manuscript. We agree with the reviewer that going to very large compressive strains may indeed lead to a subsequent decrease in T_c following the trend observed in bulk crystals, but we note that the downturn in Li et al. *National Science Review* (2025) (previously ref. 35) starts at pressures higher than about 20 GPa. Notwithstanding the challenges in direct conversion from high pressure to epitaxial strain (and the variations between reported lattice parameters at such pressures across the literature), the survey presented in Fig. R1 suggests this corresponds to in-plane pseudocubic lattice constants near 3.7 Å. In $\text{La}_3\text{Ni}_2\text{O}_7$ thin films, this equates to about ~3.46% biaxial compression, which is very challenging to stabilize in high-quality thin films [D.G Schlom et al. *MRS Bulletin* 39 (2014), D. F. Segedin et al. *Nature Communications* 14 (2023)].

We have added the following comment:

...the precise critical strain for superconductivity in $\text{La}_3\text{Ni}_2\text{O}_7$ remains to be robustly established, as does the effect of even higher compressive strains.

3. The authors frequently describe 327 as 'cuprate-like', implying a dx^2-y^2 -driven SC mechanism based on the absence of the γ band (primarily the bonding dz^2 band) in compressively strained films (as calculated in Refs. 27-28). However, even under this assumption, the Fermi surface topology of 327 remains fundamentally different from that of cuprates, as it still involves both dx^2-y^2 and dz^2 components.

Our allusions to 'cuprate-like' Fermi surface structure were motivated by the reports mentioned, but we agree with the reviewer that our previous discussion was oversimplified and have removed these descriptions.

The authors can more precisely argue that SC in thin films is unlikely to be driven by the metallization of bonding dz^2 orbitals. This is already a strong constraint on theoretical models. To support their discussion, I suggest that the authors include DFT calculation of the electronic structure based on their measured or relaxed Fmmm structure, highlighting the Fermi surface topology. They should also address the implications of the absence of a metallized γ band (i.e. absence of a hole-doped dz^2 orbital). Useful references include Phys. Rev. B 111, 014512 (2025) and the sources cited therein. Additionally, a recent ARPES study on SC 327 thin films unexpectedly reveals a metallized γ band (arXiv:2501.09255), which is confusing. The authors may comment on the discrepancy between ARPES-measured band structures and their DFT-calculated results.

We thank the reviewer for the suggestion of expanding our theoretical treatment. In addition to the calculated hopping parameters discussed above (response to #1), we have

also added novel DFT strain-decomposed calculations as Fig. 4 in the revised manuscript, along with corresponding discussion. This helps to disentangle the effects of different structural changes, such as lattice spacing and bond angle, on the final band structure. As described in the text, this analytical approach allows us to construct hypothetical lattice and electronic structures of $\text{La}_3\text{Ni}_2\text{O}_7$ which are subject to only changes in bond lengths, internal octahedral angles, or octahedral rotations identified by our ptychographic measurements. Our calculations show that band placement is predominantly affected by lattice spacing changes, while the t_{2g} hybridization is driven mostly by changes in octahedral rotation (i.e., the high- and low-symmetry bond patterns). Specifically, we observe in the compressive film a shift of d_{z^2} bands away from the Fermi surface near Gamma, consistent with previously published DFT calculations [B. Geisler et al. arXiv:2411.14600 (2024), Y. Zhao and A. S. Botana *Phys. Rev. B* 111 (2025)]. This shift in the d_{z^2} band is opposite to what is observed upon hydrostatic pressure in both bulk $\text{La}_3\text{Ni}_2\text{O}_7$ under hydrostatic pressure and in tensile strained thin films. We also observe a substantial suppression of t_{2g} hybridization common in the high symmetry SC thin film and bulk high pressure structure, but not observed in the low symmetry tensile-strained film structure. To the best of our knowledge, this is the first report of an experimentally informed and rigorous disentanglement of the octahedral degrees of freedom in this structure, made possible by a novel framework that enables exact treatment and precise tuning of these distortions.

We appreciate the reviewer's suggestion to revisit the work in Ref. [arXiv:2501.09255]. The momentum distribution curves in Fig. 3j and 3k highlight a kink-like feature traced toward the Fermi level which is presented as potentially consistent with a γ band, though we note that it is extremely difficult to identify clear evidence of a metallized γ band in the presented Fermi surface maps or single-particle dispersions of the same. Recently, a second ARPES measurement on superconducting thin films of $\text{La}_3\text{Ni}_2\text{O}_7$ grown on SLAO substrates was also reported [B. Y. Wang *et al.*, arXiv:2504.16372 (2025)] showing an *absence* of metallized γ bands in their SC films. We emphasize that both papers discuss the challenges of accurately describing their experimental results with DFT calculations. Further, the myriad challenges associated with sample synthesis, including the ozone annealing, strain relaxation, and chemical diffusion reported by other groups (note that our measurements have not revealed any such Sr diffusion in our samples), may significantly affect the electronic structure. Therefore, we believe it is premature to draw firm conclusions about the agreement or disagreement between current ARPES experiments and theoretical predictions.

4. The authors should ensure that they are not inadvertently measuring a twinned domain along the b-axis in the compressed films, as suggested in the tensile strained films by the blue region in Fig. S14. One way to verify this is by comparing the [010]-projected MEP image of tensile-strained films with Fig. 2a-b and discussing the differences. This concern arises because a slight vertical elongation of oxygen atoms in Fig. 2a-b is noticed, compared to Fig. 2c-d.

The very subtle elongation of the oxygen columns arises from relaxation in the few outer unit cells of the STEM lamella. For the high symmetry structure observed in compressively strained films grown on pseudocubic substrate, $[100]_o$ and $[010]_o$ axis projections have equivalent Ni-O bond angles as seen from the DFT relaxed structures. To verify this experimentally, we measured two orthorhombic projections orthogonal to each other via MEP. Below shows the histograms of the O-Ni-Ni angles measured along two orthogonal orthorhombic projections. Both bottom and top layer O-Ni-Ni angle values are equivalent within the error of the measurement for the high symmetry structure of $\text{La}_3\text{Ni}_2\text{O}_7$ grown on LAO and SLAO substrate.

Fig R3 | Histograms of upper and lower NiO_2 plane O-Ni-Ni angles measured along two orthogonal orthorhombic projections (here designated $[100]$ and $[010]$ for clarity, though we note that the directions are equivalent within the crystal symmetry) measured via MEP for $\text{La}_3\text{Ni}_2\text{O}_7$ thin films grown on LAO (top panel) and SLAO (bottom panel).

We have further added the text below to highlight existence of twinning in our tensile strained structure:

“We note that tensile-strained $\text{La}_3\text{Ni}_2\text{O}_7$ films show twinning in the structure (Supplemental Fig. S4). The bond angle analysis performed here on the films on NGO and STO only includes the a -axis projection while avoiding the b -axis twin.”

5. The MEP results are impressive. Could the authors provide details on the optimized parameters from Bayesian optimization used in their final reconstructions (e.g. regularization, any real-space super-sampling, point spread function, etc.) ?

The parameters used for final reconstructions through fold-slice package have been individually and carefully optimized for all >25 datasets analysed in this manuscript. The typical range of values for various optimised parameters are given in the table below. We have added this table to the Supplemental Information and are happy to provide raw datasets and their individual reconstruction parameters upon request.

Accelerating voltage	300 kV
Probe defocus	-50 : -190 Å
Convergence angle	26.3 : 30.0 mrad
Scan step size	0.43 : 0.63 Å
Slice thickness	6.0 : 9.0 Å
Z regularization	0.7 : 0.8
Outer collection angle	55 : 66 mrad
Number of scan positions	256 x 256
Detector pixels	128 x 128
Number of probe modes	8
Per pixel dwell time	100 µs

Supplemental Table S4 | Typical range of values for MEP reconstruction (via fold-slice package) parameters individually optimised for all datasets analysed in this manuscript. Z regularization is the interlayer regularization introduced in Z. Chen et al. *Science* 373 (2021).

...

Processed data and crystal structures presented in this manuscript will be made available at <https://data.paradim.org>. Raw MEP data and reconstruction parameters are available upon request to the authors.

6. In Fig. 2f, for SLAO and LAO, please consider separating the bond angles from the upper and lower layers. Presenting these as two distinct components within a histogram using different colors would improve clarity, as the current two-Gaussian fitting is not so well-defined.

We thank the reviewer for the suggestion, and have updated the O-Ni-Ni angle histograms in Fig. 2 (shown below) to separate the distortions measured in the upper and lower Ni-O₂ planes of the bilayer.

Fig 2 | (f) Ni-planar O (O-Ni-Ni) angles of all four thin films. The black solid line is a Gaussian fit to the top layer angle distribution, while the black dashed line is a Gaussian fit to the bottom layer angle distribution. The total number of O-Ni-Ni angles analyzed (N) are given for each film. (g) Average Ni-planar O angle of top and bottom layer for films under different epitaxial strains.

7. In Fig.4a-c, it appears that all \top markers indicate stacking faults associated with transitions between RP phases with $n=1$ and $n=2$. Do the authors imply that strain relaxation occurs through the intergrowth of $n=1$ RP phases? If so, please clarify this at the beginning of the section to avoid potential misunderstandings. Such defects may be detrimental instead of beneficial to SC since larger strain may not lead to better SC. Some of the statements in page 13 should be adjusted.

\top markers indicate dislocations defined by the insertion of an additional half-plane of atoms in a structure. Indeed, the reviewer correctly notes that these dislocations often

occur where changes in the stacking order due to intergrowths (such as $n=1$) are also observed. More generally, we observe multiple kinds of strain relaxation in the films including: (1) intergrowths, (2) gradual relaxation away from the interface and (3) at defects such as dislocations. We agree with the reviewer that these defects may be detrimental to SC or potentially take part in filamentary SC. We have adjusted wording of sentences during the discussion of defects to highlight both options:

“While the lattice parameters in Figs. 2 and 3 and the structures used for DFT calculations in Fig. 4 represent an ensemble understanding of the film structure, epitaxial relaxation and crystalline defects can locally modify these parameters such that they may fall outside the window required for superconductivity, hindering the global SC transition.”

“These observations also provide a tantalizing hint towards the possibility of engineering filamentary superconductivity at highly strained heterostructure interfaces or grain boundaries in bulk crystals.”

8. In recent preprints, such as arXiv: 2412.16622, resistivity of similar 327 films can change significantly during the measurements especially when the temperature is high, which was attributed to the oxygen loss. Did the authors see similar effects? For TEM measurements, is it possible to see any hints of oxygen loss during the sample preparation or sample degradation? What is the origin of the inhomogeneous oxygen stoichiometry shown in Fig. 4d? Is the O-K statistically preferred as that from the region 1 or region 2?

The statistically preferred oxygen K edge is shown below and resembles an average of regions 1 and 2 indicating mesoscale inhomogeneity in the local oxygen stoichiometry within our films. Previous measurements of bulk $\text{La}_3\text{Ni}_2\text{O}_7$ crystals have also reported local inhomogeneities in oxygen stoichiometry [Z. Dong et al. *Nature* 630 (2024)], suggesting that the variation observed in oxygen pre-peak in Extended Data Fig. E6 is likely also – at least in part – intrinsic to the as-synthesized sample.

Fig. R4 | Statistically preferred oxygen K edge measured from SC $\text{La}_3\text{Ni}_2\text{O}_7$ grown on SLAO via electron energy loss spectroscopy (EELS).

Still, we note that FIB specimen preparation can induce additional inhomogeneities, but that quantitatively measuring the effect of preparation-induced changes is extremely challenging. As discussed in E. K. Ko et al. *Nature* 638 (2025), the electronic transport behaviour of superconducting films can change over the course of several days. For the experiments performed in this manuscript, we have therefore taken extensive precautions to mitigate oxygen loss during our experiments and to try to account for any potential effect on the structural measurements reported here. The timing of our experiments is carefully coordinated such that all FIB preparation and STEM measurements are performed within 3 days of oxygen annealing. By comparison, when measurements are not as carefully time-coordinated, we observe a complete absence of the O-K edge pre-peak across the sample and inconsistent non-uniform Ni-O angles, now included as Supplemental Fig. S13 with additional comments in Methods:

Supplemental Fig. S13 | O-K edge (left) and planar O-Ni-Ni angle maps (right) measured on $\text{La}_3\text{Ni}_2\text{O}_7$ thin film on SLAO without careful time-coordination.

Minor Comments

9. Page 7: “a focused probe” may be changed to “an overfocused probe” or “a convergent probe” to avoid ambiguity.

We appreciate the suggested clarification and have updated the text accordingly:

“As in conventional high-resolution STEM imaging, a converged probe of high-energy electrons is rastered across the sample (Supplemental Fig. S1).”

10. Fig. 1c: why are the R-T curves for films on LAO and NGO only shown down to 50 K? This temperature is even higher than the onset T_c in SLAO.

The previous measurements were truncated at the onset of non-Ohmic behaviour in our electrical contacts. We have remeasured the samples down to lower temperatures and have updated Figure 1:

Fig 1 | ... c) Resistivity $\rho(T)$ of $\text{La}_3\text{Ni}_2\text{O}_7$ thin films epitaxially strained on substrates SLAO (blue), LAO (cyan), NGO (yellow), and STO (red); chemical formulas are given in the text. The nominal (bold, ϵ_{nom}) and measured epitaxial (ϵ_{exp}) strains imparted on each film are given in parentheses.

11. The manuscript refers to bond angles, but these appear to be measured as O–Ni–Ni rather than Ni–O–Ni. If so, they may be more appropriately described as “octahedral tilt angles”. Please consider revising the terminology for clarity.

We have clarified the text to refer to the measured angles as “O–Ni–Ni angle”, which more precisely reflects the measurement details. We have further provided the related Ni–O–Ni angles in the Supplementary Fig S12.

Supplemental Fig. S9 | Illustration of the measured O-Ni-Ni (left) and Ni-O-Ni (right) angles.

Supplemental Fig. S12 | a) Ni-O-Ni angles of all four thin films corresponding to the schematic at the right of Supplemental Fig. S9. The pink dashed line is a Gaussian fit the angle distribution. b) Average Ni-O-Ni angle for films under different epitaxial strains

12. Page 10: the phrase “in-plane lattice constant and octahedral symmetry as order parameters for superconductivity” is misleading. In the context of Ginzburg-Landau theory, the order parameter of a superconductor has a specific definition. Consider rewording this as “determining factors for superconductivity” instead.

We agree with the reviewer and have changed the wording accordingly.

Referee #2 (Remarks to the Author):

The authors describe a systematic structural study using a combination of STEM, MEP and XRD as a function of epitaxial strain after thin film growth on substrates with varying in-plane lattice parameters. They observe two trends as the in-plane lattice parameter approaches a -2% strain, where superconductivity is observed. The first trend is an increase of the planar Ni-O bond angles, while the 2nd weaker trend is an increase in the out-of-plane lattice parameter. These structural trends are used to conclude a "...necessity of in-plane rather than c-axis compression for achieving superconductivity." **The structural measurements are excellent and well-presented.** Before recommending for publication, there are a number of weaknesses that need to be addressed.

1) The most obvious correlation between the data is that PLD grown films are superconducting and MBE grown films are not. Please include transport and structural data on MBE-grown films on SLAO.

We thank the reviewer for raising this important point. Growing SC thin-film of $\text{La}_3\text{Ni}_2\text{O}_7$ on SLAO via MBE is currently an ongoing effort.

Recent reports, however, have already demonstrated SC MBE-grown $\text{La}_2\text{PrNi}_2\text{O}_7$ on SLAO with a SC transition near 40 K [B. Y. Wang et al. arXiv:2504.16372 (2025)] and $\text{La}_{3-x}\text{Sr}_x\text{Ni}_2\text{O}_7$ on SLAO with $T_c \sim 42\text{K}$ [B. Hao et al. arXiv:2505.12603 (2025)]. The transport and ARPES measurements reported by Wang et al. on the SC MBE and PLD films are in agreement, confirming that the overall film behavior should not be governed by the growth techniques.

We have further included comparative structure measurements of both MBE and PLD-grown thin films on LAO in Extended Data Fig. E1, showing good agreement between the qualitative pattern and quantitative measurements of bond angles and local lattice constants:

Extended Data Fig. E1 | a) Resistivity $\rho(T)$ of $\text{La}_3\text{Ni}_2\text{O}_7$ thin films epitaxially strained on LAO grown via molecular beam epitaxy (MBE) (solid line) and pulsed laser deposition (PLD) (dashed line) as described in Methods. The transport curves show similar behavior between the films grown using MBE and PLD. **b)** Ptychographic reconstruction (left) and corresponding Ni-O bond angle map (right) of a PLD-grown $\text{La}_3\text{Ni}_2\text{O}_7$ film on LAO. The Ni-O coordination observed in MBE-grown $\text{La}_3\text{Ni}_2\text{O}_7$ films on LAO (Fig. 2b) are similar to the structure observed in the films grown by PLD.

2) The trends in the transport could be discussed. There doesn't seem to be an obvious correlation between bond angle and conductivity.

Given the many complications of synthesizing $\text{La}_3\text{Ni}_2\text{O}_7$ thin films, at this stage, the transport depends not only on the intrinsic structure of the films but also on extrinsic factors such as oxygen annealing, strain relaxation, and defect concentration. This makes it difficult to draw a direct relation between the transport and bond angles.

With considerable effort in the improvement of the transport devices, however, the trend in transport is more apparent. Revised Fig. 1 shows the R vs T of newly optimized transport devices: a monotonic change in the transport behaviour from insulating (STO) to SC (SLAO) with increasing compressive strain is observed. Films on NGO and LAO with intermediate strain show metallic transport, but films on LAO under larger compressive strain are “more metallic” as exhibited by lower resistivity and lower temperature insulating upturns compared to films on NGO.

Fig 1 | ... c) Resistivity $\rho(T)$ of $\text{La}_3\text{Ni}_2\text{O}_7$ thin films epitaxially strained on substrates SLAO (blue), LAO (cyan), NGO (yellow), and STO (red); chemical formulas are given in the text. The nominal (bold, ϵ_{nom}) and measured epitaxial (ϵ_{exp}) strains imparted on each film are given in parentheses.

3) What do measurements of carrier density based on magneto transport show?

As shown in Fig. R5, the Hall coefficient measurements of SC samples indicate predominantly hole-type carriers, with a pronounced temperature dependence. This behavior could arise from the multiband nature of the system or strong electron correlations. A rough estimate of the carrier density using a simple single-band model above the SC transition yields a value on the order of $8 - 18 \times 10^{14} \text{ cm}^{-2}$ holes per NiO_2 plane. This is comparable to values found in overdoped cuprates and is also consistent with observations in Pr-doped bilayer nickelate thin films [Y. Liu et al. arXiv:2501.08022 (2025)].

[REDACTED]

4) The authors point out that the microscopy shows inhomogeneities (Fig. 4), which may lead to filamentary superconductivity in these PLD grown films. But where is the superconductivity taking place? What is needed is a discussion of the possibility that the areas with smaller out-of-plane La-La spacing are superconducting. A figure similar to Fig. 4 for the thin films grown on the other substrates would shed light on this issue.

The reviewer brings up an important discussion point. As seen in Extended Data Fig. E6 and Supplemental Fig. S2, intergrowths of $n=1$ with smaller OOP La-La spacing are observed in both compressive and tensile strained films. Robust measurements relating the compression of the c -axis lattice constant to SC without additional contributions from the other growth effects seen here, however, requires experiments beyond the scope of this work: for example, by continuously tunable strain and transport experiments in freestanding membranes or uniaxial compression cells.

As we mentioned in the response to Reviewer 1, we also note that with the recent and significant improvement in sample quality achieved through Pr substitution, we have observed a large increase in critical current density J_c . Ginzburg-Landau fitting extracted superconducting thicknesses matches well with film thicknesses extracted by Scherrer analysis of XRD and HAADF-STEM imaging. Both of these measurements point towards bulk-like superconductivity in our films rather than filamentary superconductivity [Y. Liu et al. arXiv:2501.08022 (2025)].

To highlight the increase in critical current and improvement of T_c transition with Pr substitution, we have added:

“Rare earth substitution by Pr has also demonstrated improvement in the superconducting transition [G. Zhuo et al. *Nature* 640 (2025)] and critical current density J_c [Y. Liu et al. arXiv:2501.08022 (2025)], though whether these improvements are driven by differences in ionic radii or thermodynamic stability remains to be established.”

5) How much of the film volume is relaxed and has a smaller out-of-plane lattice parameter? From the original paper reporting superconductivity, a nice XRD reciprocal space map is shown with a large volume of relaxed material diffracting with a smaller lattice parameter. (see extended data Figure 8 c, d and e in Ko, E.K., Yu, Y., Liu, Y. et al. Signatures of ambient pressure superconductivity in thin film $\text{La}_3\text{Ni}_2\text{O}_7$. *Nature* (2024). <https://doi.org/10.1038/s41586-024-08525-3>)

The superconducting sample measured in this study is equivalent to the one reported in Ko et al. *Nature* (2025) grown by the same group. Hence, in our study as well, there is a large volume of relaxed material with a smaller OOP lattice spacing. Our local characterization reveals that $n=1$ intergrowth and strain relaxation away from the interface are primary sources of the measured smaller ensemble OOP lattice parameter, as shown in the revised Extended Data Fig. E6.

6) The XRD RSM would be better plotted in a way that shows a wider dynamic range of diffracted intensity to gauge relative volume of material with various lattice parameters. The weaker diffuse diffraction in Ko et al. may imply a larger volume of relaxed material than that which is coherently strained.

We appreciate the reviewer’s suggestion. We have updated the RSM (Supplemental Fig. S7) to reflect a wider dynamic range for the MBE-grown samples on LAO, NGO, and STO. We also note that the RSMs presented in Ko et al. are from synchrotron measurements, while the RSMs in Supplemental Fig. S7 are from a lab-based diffractometer.

Supplemental Fig. S7 | Reciprocal space maps (RSM) of $\text{La}_3\text{Ni}_2\text{O}_7$ thin films with the pseudo-tetragonal (1017) Bragg peaks of $\text{La}_3\text{Ni}_2\text{O}_7$ alongside the following substrate peaks: a) (103)_{pc} pseudo-cubic peak of LAO(100), b) (332) peak of NGO(110), and c) (103) peak of STO(001). In-plane lattice constants are extracted from the $\text{La}_3\text{Ni}_2\text{O}_7$ film peaks. RSM of $\text{La}_3\text{Ni}_2\text{O}_7$ on SLAO(001) is shown in extended data of [E. K. Ko at al. *Nature* 630 (2025)]. The LAO(100) sample exhibits two substrate peaks (and thus two signals from the strained $\text{La}_3\text{Ni}_2\text{O}_7$) due to twinning in the substrate, and the left peak was used to extract the in-plane parameter based on the RSM measurement alignment.

7) Some key observations reported here have been previously published, weakening the author's case of novelty. A flattening bond angle was reported in Ref. 2 of the manuscript. Superconductivity of compressively strained thin films is reported in Ref. 6 of the manuscript. The authors' appeal to novelty is based on direct real space imaging "...for the first time-" (page 4). Discussion of points 4 and 5 would build on the novel aspects of the work.

We thank the Reviewer for allowing us to further emphasize the novelty of the manuscript. The Reviewer mentioned Ref 2, which presents XRD measurements of the bulk samples; however, such XRD measurements on few-unit-cell thin films are not possible, requiring local techniques such as electron microscopy to understand the structural distinctions and similarities between the bulk and thin film SC. As the Reviewer mentioned, Ref. 6 demonstrated the presence of SC, but also raised the crucial question of which key structural motifs unite SC in both thin films and bulk crystals. Our systematic study of structural changes across thin films of varying strain answers these questions, not addressable by other experimental probes.

Additionally, as the reviewer mentioned, STEM measurements allow for probing defects and inhomogeneity, which are washed out by bulk characterization techniques. Our thorough investigation and presentation of these inhomogeneities are not only crucial for understanding the film-scale properties of such samples reported throughout the literature, but also provide a clear road map for future efforts to achieve sharper and higher SC transitions.

We also emphasize that measurements of the lattice constants and bond angles/lengths from defect-free film regions are only possible with a local probe such as STEM. Here, we have measured the precise values of O-Ni-Ni angles, bond lengths and lattice constants to isolate the structural motifs partaking in SC and provide a correct structural starting point for *ab initio* calculations.

Building on our understanding of structural behavior in both bulk and thin films, we introduced a new DFT-based strain decomposition approach to independently analyze the impact of various octahedral distortions – namely bond length changes, internal angular modifications, and rigid rotations – on the electronic band structure (revised Fig. 4). In summary, our calculations reveal that energy shifts in the low-energy electronic structure are primarily governed by changes in octahedral bond lengths and lattice spacing. In contrast, straightening of the octahedral rotations, which gives rise to a high symmetry structure in SC thin film and bulk, results in strongly suppressing t_{2g} orbitals in the low energy "nickelate" bands. This behavior bears an intriguing resemblance to phenomena observed at the non-superconducting surface of another Ruddlesden-Popper oxide superconductor Sr_2RuO_4 (which has to-date remained largely undiscussed in the $\text{La}_3\text{Ni}_2\text{O}_7$ literature), where in-plane orbital rotations enable hybridization between the e_g and t_{2g} states. These findings suggest that even minimal e_g - t_{2g} mixing, driven by octahedral tilts or their absence, may be detrimental to SC. Beyond the implications for $\text{La}_3\text{Ni}_2\text{O}_7$ discussed here, our work demonstrates the power of this computational approach for future studies across a wide range of functional and exotic compounds, including superconductors, multiferroics, magnetic compounds, and more.

Finally, all of our measurements and theoretical models span a full strain series rather than a single compound or strain state, which is crucial to build more intuitive and predictive models of this and other systems. With this comprehensive investigation, we can draw parallels between hydrostatic pressure and biaxial strain tuning which would not be possible through observations of a single strain state. Taken together, we believe the techniques used and results presented in this manuscript are both timely and novel, providing a depth of information currently missing from $\text{La}_3\text{Ni}_2\text{O}_7$ literature.

Referee #3 (Remarks to the Author):

The manuscript Resolving Structural Origins for Superconductivity in Strain-Engineered $\text{La}_3\text{Ni}_2\text{O}_7$ Thin Films by Bhatt et al. investigates the structure of bilayer nickelate thin films, specifically on the SLAO substrate, using various experimental techniques, including MEP/STEM, XRD, transport, EELS, and ELNES. The study aims to identify structural origins of the recently observed superconducting phase by exploring different substrates spanning from compressive to tensile strain. In contrast to earlier work, the authors suggest that in-plane compression may play a more significant role than vertical compression, potentially indicating a more cuprate-like physics in bilayer nickelates than so far assumed.

The topic addressed in this manuscript is both timely and highly relevant. Given that some of the authors were involved in the discovery of superconductivity in strained bilayer nickelates, I assume the samples analyzed here are either the same as or prepared in a comparable manner to those in the original discovery paper.

However, I have concerns that the line of argument and the conclusions drawn may not be sufficiently compelling to justify publication in Nature. Additionally, while the insights presented are certainly important, of high quality, and complementary to previous studies, they may not be as groundbreaking as those in the original discovery paper.

Below, I outline several important points that need to be addressed before further consideration of the manuscript:

- Could the authors provide a hypothesis as to why, among all the studied systems, superconductivity is observed exclusively in LNO/SLAO?

We thank the reviewer for the insightful question. Based on the films studied here, it appears that the necessary compression for SC in $\text{La}_3\text{Ni}_2\text{O}_7$ falls between the strain states imposed by SLAO and LAO substrates, also illustrated by plotting reports of $T_{c,\text{onset}}$ vs in-plane lattice constant of thin films on SLAO and bulk samples from across the literature (here reproduced from Y. Liu et al. *Nature Materials* - *in press*):

[REDACTED]

With this understanding, the nominal in-plane lattice constants for films fully strained to LAO substrates ($a, b = 3.787 \text{ \AA}$) may also be enough to stabilize SC, but (as discussed in Fig. 3 of the main text) that the experimentally measured lattice constants for our films on LAO (both MBE and PLD-grown) currently fall just outside of this range due to the minor strain relaxation and defect formation which is discussed in Extended Data Fig. E6. Further improvement of the film quality and exploration of larger compressive strains will help address this point.

Furthermore, recent reports have shown that SC in thin films on LAO, NGO and STO can be stabilised through application of hydrostatic pressure [M. Osada et al. *Communications Physics* 8 (2025)]. This indicates that indeed both the compressed bond lengths and modification of bond angles influence the ability to stabilize a superconducting transition.

We have added novel DFT decomposition calculations to the paper to explore the effect of changes in crystalline symmetry, magnitudes of lattice spacing, and Ni-O buckling on the electronic band and shed light on the plausible basis of SC, now found in Figure 4 and related discussions.

- Figure 1c shows a non-monotonic evolution of resistivity as a function of strain. In particular, the STO substrate results in insulating behavior, which contrasts with recent

theoretical predictions by different groups. Could the authors comment on this discrepancy? Additionally, what structural phase is observed in films grown on STO, given that tensile strain has previously been shown to destabilize nickelate films?

Given many complications of synthesizing $\text{La}_3\text{Ni}_2\text{O}_7$ thin films, at this stage, the transport depends not only on the intrinsic structure of the films but also on extrinsic factors such as oxygen annealing, strain relaxation, and defect concentration. Hence, we caution against over-interpretation of evolution of resistivity as a function of strain.

With considerable effort in the improvement of the transport devices, however, the trend in transport is more apparent. Revised Fig. 1 shows the R vs T of newly optimized transport devices. A monotonous change in the transport behaviour from insulating (STO) to superconducting (SLAO) with increasing compressive strain is observed. Films on NGO and LAO with intermediate strain show metallic transport. LAO with larger compressive strain is identified as a “better metal” due to lower resistivity and insulating upturn at lower temperatures compared to NGO.

Fig 1 | ... c) Resistivity $\rho(T)$ of $\text{La}_3\text{Ni}_2\text{O}_7$ thin films epitaxially strained on substrates SLAO (blue), LAO (cyan), NGO (yellow), and STO (red); chemical formulas are given in the text. The nominal (bold, ϵ_{nom}) and measured epitaxial (ϵ_{exp}) strains imparted on each film are given in parentheses.

The reviewer is correct that epitaxial growth of high-quality Ruddlesden-Popper films under high tensile strain is indeed challenging, often leading to high densities of vertical rock-salt planes, as reported for example in D. F. Segedin et al *Nat Comm* 14 (2023) and T. Cui et al *Communications Materials* 5 (2024). Such defects can also create pathways

for higher oxygen mobility, potentially leading to more facile oxygen loss and hence, insulating behaviour in $\text{La}_3\text{Ni}_2\text{O}_7$. For the films studied here, we take advantage of the dynamic “floating” behavior of LaO monolayers and employ the shuttering schemes reported by J. H. Lee et al *Nature Materials* 13 (2014), Y. F. Nie et al *Nature Communications* 5 (2014) and W. Wei et al. *Phys. Rev. Materials* 7 (2023) which results in film growth with dramatic quality improvements, as evidenced by the large field-of-view STEM image in Supplemental Fig. S2 showing only minimal structural defects. This is further confirmed by our bulk x-ray measurements (2-theta thin film XRD and RSM, see Supplemental Fig. S7,8) which together confirm that the predominant structural phase is the target Ruddlesden-Popper bilayer $\text{La}_3\text{Ni}_2\text{O}_7$.

The defects observed in films grown on STO, including both structural and oxygen vacancy related, could explain the measured macroscopic insulating behaviour observed even if the system remains locally metallic as mentioned previously. We emphasize, however, that one must exercise caution when employing DFT to describe precise excited states of highly correlated materials. It is well established that standard DFT functionals systematically underestimate band gaps in semiconductors and insulators. Moreover, although DFT is formally a ground-state theory, it is common practice to interpret Kohn-Sham eigenvalues as quasiparticle energies (an approximation that fails in systems with strong electron-electron interactions). Future calculations by dynamical mean field theory (DMFT) may provide more insights in this regard, but go beyond the scope of this work. Here, we find it more instructive to focus on the structural trends which can be quantitatively reproduced between experiment and theory and to track the qualitative evolution of trends in electronic structure calculated from these models.

- The nickelate film on SLAO is grown using a different methodology compared to the other three systems, which do not exhibit superconductivity. Could the authors justify this choice and discuss in the main text whether it could influence the observed differences?

We thank the reviewer for raising this important point. Recent reports have demonstrated SC MBE-grown $\text{La}_2\text{PrNi}_2\text{O}_7$ on SLAO with a superconducting transition near 40 K [B. Y. Wang et al. arXiv:2504.16372 (2025)]. The transport and ARPES measurements of the SC MBE and PLD grown films are in agreement, confirming that the overall film behavior should not be governed by the growth techniques.

We have further included comparative structure measurements of both MBE and PLD-grown thin films on LAO in Extended Data Fig. E1, showing good agreement between the qualitative pattern and quantitative measurements of bond angles and local lattice constants:

Extended Data Fig. E1 | a) Resistivity $\rho(T)$ of $\text{La}_3\text{Ni}_2\text{O}_7$ thin films epitaxially strained on LAO grown via molecular beam epitaxy (MBE) (solid line) and pulsed laser deposition (PLD) (dashed line) as described in Methods. The transport curves show similar behavior between the films grown using MBE and PLD. **b)** Ptychographic reconstruction (left) and corresponding Ni-O bond angle map (right) of a PLD-grown $\text{La}_3\text{Ni}_2\text{O}_7$ film on LAO. The Ni-O coordination observed in MBE-grown $\text{La}_3\text{Ni}_2\text{O}_7$ films on LAO (Fig. 2b) are similar to the structure observed in the films grown by PLD.

- The authors propose that bilayer nickelates in general may exhibit physics that is more cuprate-like rather than being impacted by Ni d_{z^2} states. However, **without contributions from these states, the Fermi surface remains nickelate-like, and various nickelate systems have not exhibited superconductivity for decades.** Could the authors elaborate on what their proposed mechanism is, how it aligns with these considerations, and how it follows from their findings?

We thank the reviewer for the opportunity to expand on the work presented here. We agree with the reviewer that our discussion of ‘cuprate-like’ Fermi surface structure was oversimplified and have removed these descriptions.

To address the reviewer’s question regarding an experimentally informed understanding of SC in thin film and bulk $\text{La}_3\text{Ni}_2\text{O}_7$, we have added **novel strain-decomposed DFT calculations (Fig. 4 in revised manuscript)** to disentangle the effects of different structural changes such as lattice spacing and bond angle on the final electronic band structures of different sample geometries. As described in the text, this analytical approach allows us to construct hypothetical lattice and electronic structures of $\text{La}_3\text{Ni}_2\text{O}_7$ which are subject to only changes in bond lengths, internal octahedral angles, or octahedral rotations identified by our ptychographic measurements. Our calculations show that band placement is predominantly affected by lattice spacing changes, while the t_{2g} hybridization is driven mostly by changes in octahedral rotation (i.e., the high- and low-symmetry bond patterns). Specifically, we observe in the compressive film a shift of d_{z^2} bands away from the Fermi surface near Gamma, consistent with previously published DFT calculations [B. Geisler et al. arXiv:2411.14600 (2024), Y. Zhao and A. S. Botana *Phys. Rev. B* 111 (2025)]. This shift in the d_{z^2} band is the opposite of what is observed upon hydrostatic pressure in the

bulk $\text{La}_3\text{Ni}_2\text{O}_7$ and tensile strained thin films. We also observe a substantial suppression of t_{2g} hybridization in the high symmetry SC thin film and bulk high pressure structure compared to the low symmetry tensile-strained structure. Previous work has shown that the pairing symmetry in these compounds is highly sensitive to subtle changes in the Ni crystal field splitting [C. Xia et al. *Nat. Commun.* 16:1054 (2025)]; our results provide the basis for revisiting such calculations with more precise structural inputs. This behavior also bears an intriguing resemblance to phenomena observed at the non-superconducting surface of Sr_2RuO_4 , where in-plane orbital rotations enable hybridization between the e_g and t_{2g} states. These findings suggest that even minimal e_g - t_{2g} mixing, driven by octahedral tilts or their absence, may be detrimental to superconductivity.

We acknowledge that t_{2g} hybridization may not be the sole explanation for understanding SC in $\text{La}_3\text{Ni}_2\text{O}_7$: considerable theoretical efforts have led to other proposed driving mechanisms in realizing SC such as suppression of density waves and enhanced interlayer antiferromagnetic coupling [H. LaBollita, *Phys. Rev. Materials* 8, L111801 (2024), X. Yi et al arXiv:2505.12733 (2025)]. Future collaborative experimental and theoretical studies similar to the one presented in this manuscript will continue to help narrow the possible mechanism of SC in $\text{La}_3\text{Ni}_2\text{O}_7$.

- A significant portion of the manuscript discusses details of the oxygen relaxation under compressive strain, particularly alternating octahedral tilts versus uniform inward/outward relaxation patterns. The authors claim to observe the latter and draw parallels to high-pressure experiments. However, the energy difference between these configurations appears to be very small, based on the DFT results in the SI, which should be presented more prominently. Could the authors clarify the impact of this structural difference on the electronic properties and whether it is substantial enough to support their claims regarding its influence on superconducting pairing? Given that strain is likely the dominant effect, a clearer discussion on this aspect would be valuable to the reader.

We have expanded the theory-informed discussion of our results based on the octahedral decomposition method now presented in our manuscript, in Fig. 4 of the main text. Comparing the calculated band structures across four structural phases (bulk at low and high pressures as reported by Sun et al, 2023; biaxially strained to SLAO and STO) reveals some compelling trends: first, that the change in lattice spacings appears to drive energy shifts in the dominantly d_{z^2} band near gamma; and second, that the change in octahedral symmetry associated with high-pressure / compressive strain versus low-pressure / tensile strain results in dramatic reduction in t_{2g} hybridization from the nickel bands near E_F .

We have added the following text in the main text:

We employ this framework to study the evolution of atomic and electronic structure in $\text{La}_3\text{Ni}_2\text{O}_7$ across varying biaxial strain conditions and pressure. Starting from the non-superconducting ambient-temperature structure of bulk $\text{La}_3\text{Ni}_2\text{O}_7$ at 1.6 GPa reported by Sun et al. [H. Sun et al. *Nature* 621 (2023)] (Fig. 4b) as reference, we examine three additional geometries with all of the primary distortions and their mixings: (1) the non-

superconducting $\text{La}_3\text{Ni}_2\text{O}_7$ thin film with $\epsilon_{\text{nom}} = 1.9\%$ obtained by relaxing the bulk 1.6 GPa structure at ambient-temperature with the in-plane lattice parameters of STO (Fig. 4c); (2) the superconducting film with $\epsilon_{\text{nom}} = -2\%$ obtained by relaxing the bulk 29.5 GPa high-pressure structure at ambient- temperature from Sun et al. [H. Sun et al. *Nature* 621 (2023)] with the in-plane lattice parameters of SLAO (Fig. 4d); (3) the superconducting bulk 29.5 GPa high-pressure structure at ambient-temperature (Fig. 4e).

Atomic models and projected band structure calculations for each of the considered structures are shown in the right-most columns of Fig. 4, reproducing similar evolution reported previously, such as the movement of Ni bands near Γ (red boxes) under epitaxial strain or hydrostatic pressure [H. Sun et al. *Nature* 621 (2023), J. Yang et al. *Nature Communications* 15 (2024) , B. Geisler et al. arXiv: 2411.14600 (2024)]. Band structures for individual L- and R-distortion components are shown in the left and center columns, with expanded energy ranges and G-distortion components shown in Supplemental Figure S18. The octahedral decomposition is analytically exact: only the individual L, G, R structures are hypothetical (grey borders) while the structures in the right-most column (black borders) are by definition the DFT-relaxed structures of each sample geometry.

A stark distinction between the electronic structures of non-superconducting and superconducting sample geometries is observed upon coloring the bands by fractional local t_{2g} orbital character. Although the low-energy bands are dominantly e_g -like, we find small but non-zero t_{2g} hybridization present in both non-superconducting near-ambient bulk and tensile strained thin film structures (Fig. 4b, c). This residual t_{2g} hybridization is considerably suppressed in the two nominally superconducting -2% compressive strain thin film and high-pressure bulk structures (Fig. 4d, e). The importance of this local t_{2g} expulsion for the emergence or character of superconductivity remains to be established, though previous work has shown a sensitive dependence of the leading superconducting instability symmetry on the Ni crystal field splitting [C. Xia et al. *Nature Communications* 16 (2025)]. To-date, most calculations of superconductivity in $\text{La}_3\text{Ni}_2\text{O}_7$ have assumed fully e_g orbital character consistent with the hybridization found by our models for compressive and hydrostatic pressure, though additional insights may be provided by considering the effect of t_{2g} mixing, especially for tensile strain configurations.

Furthermore, our strain decomposition reveals that “ t_{2g} -cleaning” in the low-energy bands is mainly driven by the R distortions – that is, by the change in octahedral rotations – in combination with the electron correlations phenomenologically considered in the Hubbard U .

...

Bond length (L) and internal angle (G) distortion changes, on the other hand, preserve or even enhance few-percent t_{2g} mixing in all structures (Supplemental Fig. S18).

- The DFT calculations were performed using GGA without additional corrections, such as Hubbard terms. I suggest including DFT+U results and discussing Fig S9 more prominently in the main text to assess the robustness of the conclusions.

Structure relaxations performed within the GGA-PBE DFT framework have been shown to be the most reliable and unbiased starting point for many systems [P. Haas et al Phys. Rev. B 79 (2009)]. This approach represents a fully *ab initio* method which provides a consistent treatment of exchange and correlation across all atoms and bonding environments without relying on external tuning. While methods like DFT+*U* can improve the description of electronic correlations in systems with localized electrons, they can also artificially distort the potential energy surface and bias the relaxed structure depending on the chosen *U* value and are therefore most effective when employed in combination with high-quality experimental data of such parameters (e.g. ARPES). For structural optimizations, where the goal is to find the ground-state lattice geometry purely from first principles, it is therefore best to avoid phenomenological corrections and retain the predictive, parameter-free nature of standard GGA-PBE.

Below we include a comparison of band structure calculations based on structural relaxations of the near-ambient 1.6 GPa and bulk at 29.5 GPa structures performed in GGA, with and without including a Hubbard *U* of 4 eV on the Ni site during the relaxation and electronic structure calculation stages:

Fig. R6 | Comparison between low-energy band structures of $\text{La}_3\text{Ni}_2\text{O}_7$ projected to the global t_{2g} manifold of Ni 3d states for the near-ambient 1.6 GPa structure (upper row) and the bulk structure at 29.5 GPa reported by Sun et al. (lower row). a, d) Structure relaxations and band structures obtained within the DFT-GGA framework without Hubbard U . b, e) Hubbard $U = 4$ eV at the Ni site added for the band structure calculations. c, f) Hubbard $U = 4$ eV at the Ni site added both for the structure relaxations and band structure calculations.

The addition or omission of U during the structural relaxation yields slightly different final geometries, which in turn have noticeable effect on the energy positions and hybridizations within the low-energy electronic structure (second and third columns in Fig. R6). On the other hand, adding U during the electronic structure calculation strongly pushes the t_{2g} Ni 3d manifold towards higher binding energies (compare the first and second columns) and results in dramatic expulsion of the t_{2g} in high symmetry structures.

- Could the authors clarify the distinction between $Fm\bar{m}m$ and $I4/m\bar{m}m$ for the present $a=b$ case, particularly in relation to the phrase "lifting of crystalline symmetry"? It seems that the symmetry in the films is actually higher than that of the ambient-pressure bulk phase. Additionally, the structures shown in Fig S7 both appear tetragonal. As the authors state, temperature may also play an important role here. At what T have the structural measurements been performed?

We appreciate the chance to clarify this point. Previously, we used the designation " $Fm\bar{m}m$ -like" to indicate a similarity between the oxygen distortions observed in our samples to those in the $Fm\bar{m}m$ bulk structure reported by Sun et al (*Nature* 2023) in which all planar oxygens distort outwards from the Ni plane. As shown in Supplemental Fig. S11, the $I4/m\bar{m}m$ structure reported later by Wang et al (*JACS* 2024) has as distinct oxygen distortion, namely the inwards movement of planar oxygen. The $Fm\bar{m}m$ structure, however, also has subtly dissimilar in-plane lattice constants which prevent a formally tetragonal classification. In our thin films, however, we have taken a pseudo-tetragonal approximation $a = b$, which is why we used the term " $Fm\bar{m}m$ -like" rather than $Fm\bar{m}m$ or $I4/m\bar{m}m$.

To better avoid this confusion, we have replaced formal symmetry designations with classification of the observed oxygen distortions as either "uniform outward" (observed under compressive strain), "uniform inwards" (depicted in the $I4/m\bar{m}m$ structure reported by Wang, et al *JACS* 2024), and "alternating" (depicted in the ambient $Am\bar{m}m$ structure in bulk and also observed under tensile strain in thin films).

All the STEM experimental data discussed in the manuscript were taken at ambient temperature.

We have added the following text in the main text:

"The observed octahedral distortions in the comprehensively strained films show a close match to the originally reported structure of bulk $\text{La}_3\text{Ni}_2\text{O}_7$ under high pressure at ambient

temperatures [H. Sun et al. *Nature* 621 (2023)] in which the planar-O shift unidirectionally away from the center La-O plane towards the rock salt layer (Supplemental Fig. S11). Interestingly, the uniformly outward Ni-O bond distortions observed under compression here are distinct from uniform inward distortions reported by more recent bulk refinements [L. Wang et al. *Journal of the American Chemical Society* 146 (2024)] under high pressure at low temperature conditions (Supplemental Fig. S11). Whether this inversion in the O distortion direction corresponds to a temperature-dependent structural transition under pressure, and whether the superconducting thin film undergoes a similar low-temperature structural transition remain open questions, the latter going beyond the current limitations of cryogenic electron microscopy [E. Bianco and L. F. Kourkoutis *Accounts of Chemical Research* 54 (2021)].”

- I assume that the colors in Figure 2a represent the projection of horizontal Ni-O bond angles onto a plane perpendicular to the viewing axis, rather than being related to rotations about the vertical c-axis. Could the authors confirm this interpretation and provide additional details on the representativeness of the images shown, particularly since the authors mention later that defects and interface roughness strongly impact these structural aspects.

The reviewer is correct that the colors in Fig. 2 represent the projection of the Ni-planar O angle relative to the neighboring Ni plane. We have added a detailed explanation of how we have measured the angle in the manuscript to remove any confusion:

“We quantify local Ni-planar O angles by first obtaining Ni and planar O positions through a 7-parameter Gaussian fitting of the atomic columns in MEP reconstructions. Using the measured positions, we then obtain the projected Ni-planar O bond angle by measuring the in-plane O-Ni angle with respect to the horizontal Ni plane as illustrated in Supplemental Fig. S9, giving the O-Ni-Ni angle.”

The statistical analysis presented as histograms in Fig. 2f considers several reconstructions from many regions of each film to build a representative sampling. The total number of bonds N analyzed for each film is listed below each histogram.

- To improve clarity, reference spectra should be added to Figure 4d to better support the interpretation of the prepeak and its relation to oxygen concentration.

We thank the reviewer for the suggestion and have added a reference spectrum of the O-K edge in bulk $\text{La}_3\text{Ni}_2\text{O}_{7-\delta}$ with $\delta \sim 0.04$ from Z. Dong et al. *Nature* 630 (2024) to illustrate the pre-peak dependence on oxygen concentration.

Extended Data Fig. E6 | Impact of local inhomogeneities on lattice and oxygen structure. a) Large field-of-view ADF-STEM image of the $\text{La}_3\text{Ni}_2\text{O}_7$ film on SLAO. Dashed lines mark the interface between film and substrate. Yellow Ts mark crystalline dislocations. b-c) In-plane (b) and out-of-plane La-La spacing (c) measured by quantitative atom-tracking in the region marked by white box in (a). Orange arrows mark relaxation near defects. Average IP and OOP La-La spacings in clean regions of the film are marked by black arrows on the colorbar. d-e) O-K edge ELNES (d) and Ni-O bond angle map (e) from three different regions of the same $\text{La}_3\text{Ni}_2\text{O}_7$ film on SLAO (ADF images of regions 1 and 2 are shown in Supplemental Fig. S5). Dotted pink line in (d) reproduces the O-K edge measured in bulk $\text{La}_3\text{Ni}_2\text{O}_{7-\delta}$ with $\delta \sim 0.04$ reported in Z. Dong et al. *Nature* 630 (2024).

Referee #1 (Remarks to the Author):

The authors have done an **exceptional job in revising the manuscript, thoroughly addressing all previous comments and significantly strengthening the work with new DFT calculations**. The octahedral distortion-decomposed DFT analysis is a **particularly innovative and impactful addition, providing profound insights** into the specific structural degrees of freedom that govern superconductivity. This approach **convincingly underscores the critical importance of precise local structural measurements**, elevating the interest and relevance of this work to a broad scientific audience. I am pleased to recommend this excellent manuscript for publication in Nature.

I have a few minor comments and suggestions that could further enhance the clarity and impact of the manuscript:

1. The new DFT calculations exploring the independent effects of bond length (L) and octahedral distortions (R , G) are highly valuable. However, the description of the chosen parameters could be clearer.
 - 1). Please explicitly state what the length L represents (e.g., the OOP or IP Ni-O bond length) and specify the magnitude of change applied in the calculations (e.g., $\pm 2\%$, $\pm 4\%$). If Extended Data Fig. E3 reads like that L changes up to -4% (~ 8 pm), then it is much larger than the measured value.

We appreciate the reviewer's suggestion to improve the clarity of our analysis. Here, L is not a scalar but rather a three-dimensional matrix which represents the full tensor of arm-length changes from the near-ambient octahedra to those in our DFT relaxations under the various strain or pressure conditions as indicated for each row. The first column of Extended Data Figure 3 presents the total octahedral strain tensor describing the transformation of octahedra from the near-ambient structure to the three strain and pressure states investigated here: tensile strain (STO), compressive strain (SLAO), and high pressure. The three columns L , G , and R depict a further decomposition of the total strain tensor into the distortion components corresponding to octahedral arm lengths (L), internal angles (G), and rigid rotations (R). Note that ε_{zz} and L_{zz} , etc. refer only to changes in the octahedra contained within a single bilayer rather than changes to the unit cell. In the bilayer structures, the outer apical oxygens relax into the rock salt layer which induces a non-centrosymmetry in the octahedra along z , which we account for in our description of the generalized octahedral parameterization in Supplemental Information Section IX. This apical distortion induces nontrivial F matrices which, together with particular combinations of the L , G , and R matrices, allow us to calculate the virtual structures. The L_{zz} value mentioned by the reviewer is based on models of the structure under hydrostatic pressure (Extended Data Fig. E3c second column), $P = 1.6$ and 29.5 GPa previously reported by Sun et al. *Nature* (2023) rather than the films measured here.

It is important to note that these component matrices combined generate the final structure probed experimentally via STEM and do not individually correspond to any experimentally measured (or measurable) values, though Supplemental Figure S19 provides visualizations and values of the mathematical parameters required to fully describe the final-state octahedra of the

four pressure/strain conditions discussed in detail in Figure 4. For direct comparison with our measurements from STEM data, we have also tabulated projected values from the DFT-relaxed structures in Supplemental Information Table S3 (see below). We have further clarified this point in the main manuscript:

“Although the component matrices do not individually correspond to any experimentally measured (or measurable) values, such comparisons can be derived from the complete octahedral parameters presented in Supplemental Figure S19. For ease of comparison, we also tabulate in Supplemental Table S3 structural parameters for strained $\text{La}_3\text{Ni}_2\text{O}_7$ calculated from our DFT relaxations and projected for direct analogy to STEM measurements.”

2). It would be beneficial to discuss how these computationally tested values compare to the experimentally measured changes. For instance, the experimental OOP Ni-O bond length change between tensile and compressive strains is relatively small (~ 2 pm), whereas a -4% change in L represents a much larger distortion (~ 8 pm). This OOP bond length change is much smaller than c -axis change (40 pm, 2.5%). This may explain why the expansion along c -axis is not very important since the Ni-O bond length does not change obviously (only 2 pm).

The reviewer raises an interesting question regarding the distinction between c -axis spacing and OOP Ni-O bonds lengths. Experimentally, we find the change in c -axis lattice parameter between SLAO and STO thin films is ~ 0.28 Å, while the change in Ni-apical O is ~ 0.04 Å, as listed in Supplemental Table S2. For 8 vertical Ni-O bonds in one unit cell, this corresponds to ~ 0.32 Å change along the c -axis lattice direction, in good agreement with the total change in the c -axis within the bounds of experimental error, and consistent with our theoretical models shown in Supplemental Table S3 and Supplemental Figure S19.

Supplemental Information Tables S2 and S3 list structural parameters experimentally measured by MEP or STEM and those from our DFT relaxations. To best facilitate direct comparison with STEM-measured values, we have updated Table S3 to provide the distances measured along the same projections as our experiments:

Substrate (nominal ϵ)	SLAO (-2.0%)	STO (+1.9%)
Projected Ni-planar O (O-Ni-Ni) angle, top layer ($^\circ$)	1.2	7.2, -7.3
Projected Ni-planar O (O-Ni-Ni) angle, bottom layer ($^\circ$)	-1.2	-7.2, 7.3
Projected Ni-planar O-Ni angle ($^\circ$)	177.6	165.6, 165.5
In-plane axis (Å)	5.31	5.52
c (Å)	20.812	19.986

Projected OOP Ni-outer apical O spacing (Å)	2.31	2.14
Projected OOP Ni-inner apical O spacing (Å)	1.97	1.95
Volume $a \times a \times c$ (Å ³)	587.2	609.5

Supplemental Table S3 | Structural parameters for strained La₃Ni₂O₇ calculated from DFT relaxations as projected for direct comparison to STEM measurements. Full crystal structure models are provided as Supplemental Data.

The theoretical values are in generally good agreement with the experimental data and follow the same overall trends. As we note in Figure 1, the experimentally measured films are slightly relaxed from nominal strain states, evidenced by lattice parameters and bond lengths which are closer to the ambient conditions than those predicted by fully strained DFT models.

3). Similarly, a brief note on whether the magnitudes of the R and G distortions used in the DFT models are consistent with the experimentally observed ranges would be helpful.

As discussed above, the *L*, *G*, *R* decompositions are a purely analytical tool rather than experimental observables. The most appropriate comparisons between experiment and theory are the structural parameters which we measure by STEM and their comparison to our DFT relaxations in Supplemental Information Tables S2 and (updated) S3.

2. The statistical analyses of O-Ni-Ni angles in Fig. 2f could be further improved. The two distinct oxygen sites within the NiO₂ plane (i.e., even and odd sites) can be treated as separate sublattices, since the neighboring two angles along the plane are not always equal and measured separately. This could further statistically verify the unidirectional alignment and the symmetry change in the compressive films.

We appreciate the reviewer's suggestion for further dividing the bilayer into odd and even sites for Ni-O bond analysis. Below, we show histograms of Ni-O angles measured from MEP reconstructions of LNO films grown on SLAO and STO, now divided not only between the top and bottom layer of the bilayer (as in Fig. 2f) but also between odd and even sites. The odd and even sites for LNO/SLAO under compressive strain show equivalent angles, while on STO, the angles alternate between positive and negative values for neighbouring sites. Consistent with Fig. 2, this clearly shows the lower (higher) symmetry structure of tensile (compressive) films. We have added the following figure to the Supplemental Information and corresponding reference in the main text:

Supplemental Fig. S15 | Histograms of upper and lower NiO_2 plane O-Ni-Ni angles separated between odd and even sites measured via MEP for $\text{La}_3\text{Ni}_2\text{O}_7$ thin films grown on SLAO (top panel) and STO (bottom panel). As discussed in Figure 2 of the main text, the compressively strained films exhibit distributions for the bottom and top layer distortions which can be captured by a single Gaussian, while the tensile strained films require two-Gaussian fits to fully capture the positive and negative distortions in each layer.

“... unidirectionally in the compressive films but alternate in tensile films, as also clearly exhibited in O-Ni-Ni angle histograms separated between the alternating sites in Supplemental Fig. S15.”

3. Supplemental Table S4 effectively outlines the parameter ranges for the Bayesian optimization. For completeness and transparency, it would be useful to also state the final, optimized values for key parameters (such as convergence angle and scan step) that resulted from this process. A note on whether these values converged consistently across different datasets would further demonstrate the robustness of the methodology.

We have added the table below to Supplemental Information with exact values of parameters used for the reconstruction of the datasets measured on different days.

	Fig 2a	Fig 2b	Fig 2c	Fig 2d	Fig E1	Fig E2	Fig E6
Accelerating voltage (kV)	300	300	300	300	300	300	300
Probe defocus (Å)	-81	-97	-75	-177	-144	-70	-94
Convergence angle (mrad)	27.6	27.3	27.4	27.8	27.5	28.0	27.4
Scan step size (Å)	0.44	0.43	0.43	0.43	0.43	0.41	0.43

Supplemental Table S5 | Values for MEP reconstruction parameters individually optimized.

4. The reference in the caption of Fig. S11 is not correctly displayed.

We thank the Reviewer for their careful reading of our work and have fixed the reference.

Referee #2 (Remarks to the Author):

The authors have improved the manuscript with new experiments, but there are still critical gaps. In particular, there are some issues that need to be addressed.

The first is less than ideal experimental design, and it seems like the MEP imaging, according to the authors, would best be done on carefully prepared $\text{La}_2\text{PrNi}_2\text{O}_7$ as a function of annealing and strain.

We agree with the reviewer that a careful study of optimized $\text{La}_2\text{PrNi}_2\text{O}_7$ (LPNO), along with other rare-earth (RE) substitutions, as a function of annealing and strain is relevant. It is equally both important and valid, however, to understand the extensively explored parent compound without any substitution. Currently, there is no clear experimental and theoretical understanding of the consequences of substitution, and as the original discovery LNO still makes up the bulk of experimental literature. Furthermore, accounting for A-site cation disorder induced through RE substitution in *ab initio* calculations is significantly more complex and will necessarily rely on simplifying assumptions. Un-substituted LNO across varying strain provides an ideal and robust system for uniting the understanding of superconductivity (SC) in bulk and thin films and for bridging experimental measurements and first principles calculations.

In line with the reviewer's suggestions, however, we have also been working to understand the effects of Pr-substitution on the structural and electronic properties. The atomic resolution STEM experiments on LPNO films are *significantly* more challenging due to increased sensitivity during STEM sample preparation, requiring the development of a modified characterization pipeline which falls beyond the scope of this manuscript.

As pointed out in initial comments, MBE films grown for this work are not superconducting while PLD films are. Referencing other groups' work is not sufficient, as specific growth and processing conditions used in those works may lead to differing strain states and superconductivity. Also, do the Pr based MBE-grown films exhibit the same progression of structure observed using multislice electron ptychography (MEP)? Perhaps there may be two essential ingredients for realizing superconductivity in the RP nickelates, or just one not captured by the present MEP study.

We have been unable to grow SC MBE films for our study due to difficulty in obtaining high-quality SLAO substrate since the explosion of interest in RP nickelate thin films on SLAO. We again emphasize that direct comparison between MBE and PLD grown LNO/LAO films provided in the previous round of review shows remarkable agreement in both structural and electronic properties between the two growth methods (Review Fig. R1). Although the Reviewer is correct that "specific growth and processing conditions used in those works may lead to differing strain states", here we have directly measured and reported both the average and local lattice constants in all of our films to circumvent such ambiguities.

Fig. R1 | a) Resistivity $\rho(T)$ of $\text{La}_3\text{Ni}_2\text{O}_7$ thin films epitaxially strained on LAO grown via molecular beam epitaxy (MBE) (solid line) and pulsed laser deposition (PLD) (dashed line) as described in Methods. The transport curves show similar behavior between the films grown using MBE and PLD. Ni-O bond angle map of MBE-grown (left) and PLD-grown (right) $\text{La}_3\text{Ni}_2\text{O}_7$ film on LAO. The Ni-O coordination observed in MBE-grown $\text{La}_3\text{Ni}_2\text{O}_7$ films on LAO are similar to the structure observed in the films grown by PLD.

We further point to a study done by Wang et al. comparing the ARPES data of SC films grown on SLAO by MBE and PLD that directly addresses the concern raised by the reviewer [B. Y. Wang et al. *arXiv:2504.16372* (2025)], in which they observe no differences in the electronic structure arising from the growth technique.

[REDACTED]

From these reports, along with the fact that several studies from different groups have demonstrated superconductivity in compressively strained films grown by PLD [R1-R7] and MBE [R8-R10] (references given below), we find no strong evidence to suggest intrinsic differences in structure or electronic properties between films grown via different methods beyond the normal lab-to-lab variations of stoichiometric control or in oxygen annealing procedures. We agree with the reviewer that a comprehensive and systematic investigation of the myriad synthesis tuning parameters in these compounds (e.g., ozone annealing time) is of great interest to the field, but goes beyond the scope of our study regarding the relationship between atomic-scale distortions and epitaxial strain.

[R1] M. Osada et al. *Communications Physics* **8**, 251 (2025)

[R2] E. K. Ko et al. *Nature* **638**, 935–940 (2025)

[R3] L. Xiang et al. *arXiv:2508.11581* (2025)

[R4] Y. Tarn et al. *arXiv:2510.27613* (2025)

[R5] Y. Shi et al. *Adv. Mater.* e10394 (2025)

[R6] Y. Liu et al. *Nature Material* **24**, 1221–1227 (2025)

[R7] Q. Li et al. *arXiv:2507.10399* (2025)

[R8] M. Wang et al. *arXiv:2508.15284* (2025)

[R9] B. Hao et al. *Nature Materials* **24**, 1756–1762 (2025)

[R10] B. Y. Wang et al. *arXiv:2504.16372* (2025)

A more optimal experimental design would be to investigate the structure using MEP and XRD of the $\text{La}_2\text{PrNi}_2\text{O}_7$ s as a function of the ozone annealing done by Y. Liu et al, *Nature Materials* volume 24, pages1221–1227 (2025). (Ref 11 in the MS).

We appreciate the Reviewer for suggesting additional experiments beyond those already reported elsewhere. Recent x-ray measurements in bilayer thin films [D. F. Segedin et al. *arXiv:2506.10262* (2025)] and atomic-resolution STEM measurements in bilayer bulk systems [Z. Dong et al. *Nature Materials* **24**, 1927–1934 (2025); X. Chen et al. *Adv. Mater.* **37**, e04238 (2025)] have demonstrated the presence of distinct structures under excess oxygen conditions, prompting the need for a thorough, systematic study of the structural evolution as a function of ozone annealing. As mentioned in the previous comment, however, systematic atomic-resolution characterization via MEP is not trivial for LPNO thin films. The suggested experiment is underway but falls beyond the scope of this manuscript, which investigates the impact of strain rather than oxygen stoichiometry. We hope that our detailed work here will present a useful roadmap for other groups to perform equally thorough investigations.

An improved XRD analysis of the fraction of film coherently strained as well as synchrotron measurements of octahedral rotations would strengthen the conclusions. Contrary to the author's assertion that such measurements are not possible, they are now done at a number of US and international beamlines (cf, *X-Ray Scattering Techniques for Epitaxial Oxide Thin Films*, in "Measuring Octahedral Rotations in Perovskite Oxide Heterostructures" by S. J. May, Springer 2025). As the authors point out in the rebuttal, the use of synchrotron diffraction would be superior to lab-based diffraction. The salience of this point is indicated by the improved presentation of XRD for 327 La film on LaAlO_3 where a large volume of diffuse intensity is observed, leading one to surmise that a major fraction of the film is not coherently strained to the substrate. If so, are the MEP structural parameters intrinsic to the RP nickelates or an exceptional property of the film?

The reviewer brings up an important question regarding the need for a local probe of MEP to quantitatively analyze the structure of bilayer nickelates and its complementarity to macroscopic probes such as synchrotron measurements.

We first address the representativeness of our STEM results:

As electron microscopists, we are keenly aware of the limitations of STEM to probe very small length scales with respect to the starting material and thus take great care to ensure that our data is representative of overall trends to the best of our understanding. More explicitly:

The MEP bond angle and bond length analyses are based on **40 distinct datasets** with **each dataset spanning a unique field of view (FOV) of ~10nm x 10nm**, corresponding to a total of 2082 (SLAO), 5235 (LAO), 1817 (NGO) and 825 (STO) bonds in the LNO films analyzed, as indicated in Fig. 2f. We have further added Supplemental Fig. S3 that includes MEP reconstructions from randomly chosen regions of LNO films grown on SLAO and NGO showing

clear agreement of the respective structure across the samples. Along with this, the ADF-STEM analysis of lattice spacing includes **23 unique datasets** from 23 different locations on the samples. The precise locations of atomic resolution measurements (both ADF-STEM and MEP) are chosen essentially at random aside from avoiding fields of view which are dominated by extended defects present in some films, that are analyzed separately in Supplemental Information section III and elsewhere [E. K. Ko et al. *Nature* 638, 935–940 (2025), Y. Liu et al. *Nature Material* 24, 1221–1227 (2025)]. In our continued investigations over the past 1.5 years, we have not observed inconsistency or deviations from the results succinctly presented here. Furthermore, the experimental results are in generally good agreement with both mesoscopic measurements by XRD as well as theoretical DFT relaxations. We are therefore confident that MEP and ADF structural parameters measured in this work are intrinsic to RP nickelate thin films. We would also like to emphasize the widely accepted importance and relevance of local measurements when conducted, analyzed, and presented responsibly: for example, conclusions from oxygen vacancy statistics in bulk crystals of $\text{La}_3\text{Ni}_2\text{O}_7$ by Dong et al. [*Nature* 630, 847-852 (2024)] were based on similarly local measurements sampling ~ 1000 atomic columns from just three distinct $10 \times 10 \text{ nm}^2$ fields of view.

Supplemental Fig. S3 | Multislice electron ptychography reconstructions from randomly chosen regions of $\text{La}_3\text{Ni}_2\text{O}_7$ grown on SLAO (left) and NGO (right) showing agreement between the datasets. Scale bar is 5 Å.

We further point out the limitations of the XRD measurements suggested by the reviewer:

Quantitatively estimating the strained fraction from XRD RSM is not possible without extensive simulation to account for relaxed regions which inherently scatter with a different structure factor compared to strained regions. We have already noted different avenues of relaxation present in the films (Supplemental Information Section III, Supplemental Figure S4). Our detailed analysis is focused on the coherent $\text{La}_3\text{Ni}_2\text{O}_7$ nickelate regions as described in the manuscript. Furthermore, we discuss these results within the context of reported mesoscale (x-ray) measurements and find generally good agreement. Taken all together, we are confident that the MEP work presented is intrinsic to the RP nickelates.

The x-ray technique discussed in the work by S. J. May provides an average value of atomic positions throughout the film. However, precisely and accurately resolving the oxygen sublattice – a key advantage of MEP over other microscopy methods as discussed in the manuscript – is challenging with x-rays due to weak scattering from the light oxygen anions. Although x-ray could characterize the presence (or absence) of octahedral rotations via half-order reflections, any reliable quantitative values such as bond angles are challenging to obtain, particularly if the sample is not perfectly homogeneous: x-ray measurements average over both the beam size and \sim micron depths or more. In our work, we directly observe from ADF-STEM characterization that LNO films exhibit lateral inhomogeneity on a *few-nanometer* length scale. As a result, the refined structure from synchrotron measurements will necessarily be a convolution of defect regions and pristine bilayer regions. We would further like to emphasize that these inhomogeneities include variations in the Ruddlesden-Popper stacking (as illustrated, for example, in Supplemental Fig. S4, S5, also characterized in E. K. Ko et al. *Nature* **638**, 935–940 (2025), Y. Liu et al. *Nature Material* **24**, 1221–1227 (2025), other works), which significantly distort any quantitative analysis of octahedral rotations via non-local techniques such as synchrotron x-ray. For example, the $n=1$ (La_2NiO_4) bulk compound does not intrinsically host octahedral rotations, so inclusions of $n=1$ would likely manifest as artificially “suppressed” rotations in the x-ray measurement, obscuring the structure in $n=2$ bilayer regions of interest. Along with the stacking variations, we also observe phase slips in the oxygen sublattice itself. The depth resolution afforded by our MEP technique enables us to probe this directly: Supplemental Fig. S2 shows two slices from different z depths of the same MEP reconstruction of an LNO film under tensile strain on STO. Although the cation (La, Ni) lattice is fully ordered in this region, slices from the upper and lower half of the STEM lamella thickness show opposite displacements of the O atomic columns. Averaging over the full projection (thickness \sim 28 nm) effectively washes out any ability to resolve the O displacements – x-ray measurements which average over much larger scale, up to microns or more, would similarly lose such information. Thus, MEP is uniquely suited to extracting quantitative information of bond angles and lattice constants from the pristine parts of these films, especially given the care we have taken to ensure unbiased statistical sampling.

To address the importance of these considerations, we have added the following figure and text to the Supplemental Information and Methods section, respectively:

Supplemental Fig. S2 | a) (Left) Schematic of depth information provided through multislice electron ptychography. (Right) Two depth slices separated by ~ 10 nm obtained from the same ptychographic reconstruction of LNO/STO. Areas highlighted by white squares are enlarged in b) showing opposite displacements of oxygen columns. When averaged through the total lamella depth (top), the oxygen column position can no longer be precisely identified and the difference between the two regions is lost.

“The MEP bond angle and bond length analyses in this manuscript are based on 40 distinct datasets with each dataset spanning a unique field of view (FOV) of ~ 10 nm \times 10 nm, corresponding to a total of 2082 (SLAO), 5235 (LAO), 1817 (NGO) and 825 (STO) bonds analyzed, as indicated in Fig. 2f. The ADF-STEM analysis of lattice spacing includes 23 unique datasets, each spanning a total of ~ 30 nm \times 30 nm FOV.”

The authors appear to assume a particular conclusion by stating that “Here, we have measured the precise values of O-Ni-Ni angles, bond lengths and lattice constants to isolate the structural motifs partaking in SC and provide a correct structural starting point for ab initio calculations.” Electron microscopy does not identify the SC regions.

The Reviewer is correct that electron microscopy cannot identify the regions of the sample which undergo a superconducting transition. Nonetheless, we would like to emphasize that the high-quality SC thin films presented here and elsewhere [L. Xiang et al. *arXiv:2508.11581* (2025); Y. Shi et al. *Adv. Mater.* e10394 (2025); B. Hao et al. *Nature Materials* **24**, 1756–1762 (2025)] exhibit predominantly bilayer structure and have been found to demonstrate bulk-like superconductivity [Y. Liu et al. *Nature Material* **24**, 1221–1227 (2025)]. From these observations, we believe it is reasonable to assume that superconductivity is indeed occurring in the pristine bilayer regions of the thin films and not in the defect regions that occupy a fraction of the total film volume. Here, we have measured the precise values of O-Ni-Ni angles, bond lengths, and lattice constants in such pristine regions of the thin films while highlighting the defects present. To explicitly acknowledge the limitation of electron microscopy in identifying superconducting regions, we have added:

“Although STEM techniques can not directly identify superconducting regions within the film, high-quality thin films [Y. Liu et al. *Nature Materials* **24**, 1221–1227 (2025), L. Xiang et al. *arXiv:2508.11581* (2025); Y. Shi et al. *Adv. Mater.* e10394 (2025); B. Hao et al. *Nature Materials* **24**, 1756–1762 (2025)] have been found to exhibit predominantly bilayer structure and demonstrate bulk-like superconductivity [Y. Liu et al. *Nature Materials* **24**, 1221–1227 (2025)], suggesting that the pristine bilayer regions examined in Figs. 2 and 3 and used for DFT calculations likely host superconductivity.”

The inclusion of Hall measurements is incomplete. In the spirit of the experimental approach, measurements are best presented as a function of strain to eliminate strain induced doping effects as the dominant change with degree of strain.

As the reviewer requested, Fig. R2 shows the Hall coefficient measurements of the full strain series. We again note remarkable agreement between PLD and MBE-grown films on LAO.

Fig. R2 | a) Hall coefficient measurements of superconducting $\text{La}_3\text{Ni}_2\text{O}_7$ thin films grown on SLAO substrates. The blue dots are data taken from E. K. Ko et al. *Nature* 638 (2025), green and red dots are data obtained from additional samples. b) Hall coefficient measurements of $\text{La}_3\text{Ni}_2\text{O}_7$ thin films grown on LAO substrates via MBE (blue) and PLD (violet) growth methods. Hall coefficient measurements of $\text{La}_3\text{Ni}_2\text{O}_7$ thin films c) grown on NGO substrates and d) grown on STO substrates.

Still, however, we caution against overinterpretation of the Hall coefficient due to its strong dependence on other conditions, such as oxygen content in the film. This is exemplified through the large variation of Hall coefficients reported in the literature and demonstrated systematically by M. Wang et al. *arXiv:2508.15284* (2025). A more comprehensive investigation of these effects specific to Hall measurement must be undertaken before conclusions regarding doping effects can be deduced from such data.

The addition of a new section on first principles theory is helpful, but raises a technical question. The authors need to justify in more depth for this particular nickelate system the validity of using first principles theory to predict the electronic structure via the approach of artificially moving the atoms away from a ground state structure, the “virtual structures” on page 12.

We appreciate the reviewer for indicating clarification needs. The ground state structures under biaxial strain and hydrostatic pressure and their calculated electronic bands (right-most column of Fig. 4b-e) are a result of coupled distortions such as change in bond lengths, rigid rotations of octahedra etc. Determining which distortion in particular drives the key electronic change is not possible experimentally due to the coupling between them in real systems. Here, we are using the virtual structure calculations in which each structure embodies a specific distortion to qualitatively separate the effects of different experimentally observed distortions on the electronic bands. We further emphasize that the discussion guided by our analysis of octahedral distortions is a qualitative rather than quantitative one: we are not claiming to predict precise energy scales or band structures, but rather to provide systematic insight into understanding how certain distortions modify aspects of orbital character under different strains. This technique has been previously used to successfully understand the ARPES data in Sr_2RuO_4 [E. A. Morales et al. *Phys. Rev. Lett.* 130, 096401 (2023)] with similar octahedra distortions to LNO, providing a clear precedent for this approach. We include extensive discussion of the assumptions and limitations of our model calculations, including considered structure extent, Hubbard U term etc. in Supplemental Information Sec. IX.

We have made changes to the manuscript to further emphasize qualitiveness of this technique:

“To **qualitatively** disentangle the impact of structural parameters characterized in Figs. 2 and 3, we leverage a theoretical framework for modeling distortions in two-dimensional systems of corner-sharing octahedra such as those within each bilayer of the $\text{La}_3\text{Ni}_2\text{O}_7$ structure.”

“We employ this framework to qualitatively study the evolution of atomic and electronic structure
...”

In its current form, publication in Nature is not recommended.

Referee comments in black

Our reply in blue

Changes to the text in green

Referee #1 (Remarks to the Author):

The authors have adequately addressed the concerns, and the manuscript is now acceptable. I have one additional technical question regarding Supplemental Table S5: the convergence angle and scan step size differ across the listed datasets. The difference is as large as $\pm 1\%$ for convergence angle, which is significant for MEP. Could the authors clarify whether this variation stems from optimization errors or from intentional adjustments to the imaging conditions for each dataset?

Variations in convergence angle across different datasets taken on different days is a common occurrence in ptychographic reconstruction and is attributed to the precision of optimization due to coupled parameters. In ptychographic reconstruction, different optimization parameters such as convergence angle, step size, and defocus are interlinked; adjustment of one can lead to a change in the other. The variation observed here is within normal range and is also reported in other papers [H. KP et al. *Nature Materials* 24, 1433–1440 (2025), S. Karapetyan et al. *Nature Communications* (2026)]. We have added this discussion to the Supplemental Information Section II:

“Note that different optimization parameters within each reconstruction, such as convergence angle and step size are interlinked; the variation in value for convergence angle shown here are within a typical spread previously reported [H. KP et al. *Nature Materials* 24, 1433–1440 (2025), S. Karapetyan et al. *Nature Communications* (2026)].”

Referee #2 (Remarks to the Author):

A critical weakness in the manuscript remains, which is the lack of superconductivity in MBE films grown by the team at any strain state. They need to perform the reference check that superconductivity in their MBE films occurs, in an overlapping strain range with superconducting PLD films, or the natural Occam’s razor conclusion is that their MBE films are not superconducting while the PLD films are. A delay in substrate delivery is not uncommon in the field, and for an impactful study is worth the wait. Once the baseline demonstration is made, the manuscript is in much stronger position for publication.

We thank the reviewer for suggesting avenues to further strengthen the manuscript. As elaborated in the previous review responses, we do not currently have superconducting MBE-grown films, but many other studies [R1-R3] have already demonstrated superconductivity in MBE-grown $\text{La}_3\text{Ni}_2\text{O}_7$ films on SLAO with an overlapping strain range to what we and others [R4-R10] have reported from PLD growth. Wang et al.’s ARPES results on SC films grown on SLAO by MBE and PLD show no considerable differences, indicating a lack of evidence to suggest intrinsic

differences in structure or electronic properties between films grown via different methods beyond the normal lab-to-lab variations of stoichiometric control or in oxygen annealing procedures.

- [R1] M. Wang et al. *Phys. Rev. Lett.* 136, 066002 (2026)
- [R2] B. Hao et al. *Nature Materials* 24, 1756–1762 (2025)
- [R3] B. Y. Wang et al. *arXiv:2504.16372* (2025)
- [R4] M. Osada et al. *Communications Physics* 8, 251 (2025)
- [R5] E. K. Ko et al. *Nature* 638, 935–940 (2025)
- [R6] L. Xiang et al. *arXiv:2508.11581* (2025)
- [R7] Y. Tarn et al. *arXiv:2510.27613* (2025)
- [R8] Y. Shi et al. *Adv. Mater.* e10394 (2025)
- [R9] Y. Liu et al. *Nature Material* 24, 1221–1227 (2025)
- [R10] Q. Li et al. *arXiv:2507.10399* (2025)

Still, we appreciate the necessity for precision in this regard and have expanded our discussion of MBE vs PLD in relation to film properties:

“To-date, superconductivity has been demonstrated in strained bilayer nickelate thin films grown by pulsed laser deposition (PLD) [R4-R10], molecular beam epitaxy (MBE) [R1-R3], and gigantic oxidative atomic layer-by-layer epitaxy [G. Zhou et al. *Nature* 640, 641–646 (2025)]. In this work, the films on STO, NGO, and LAO are grown by MBE, while the superconducting film on SLAO is grown by PLD. Though we cannot definitively exclude the possibility that some properties such as superconductivity may be sensitive to the growth method or other details of film synthesis, comparison between electronic transport and atomic structure of MBE- and PLD-grown films on LAO finds good agreement between both techniques (Extended Data Fig. E1). These observations are in line with angle-resolved photoemission experiments which found no distinction between the electronic structure measured in MBE or PLD-grown superconducting films [R3].”

Beyond this key issue, the authors have made meaningful improvements to the manuscript in response to earlier comments, and the study addresses a timely problem in the field of strained nickelate thin films and superconductivity. Some aspects of the structural analysis would benefit from further clarification to bolster the overall conclusions.

We appreciate the reviewer for recognizing the impact of this study in the field of strained nickelate thin films and superconductivity.

One point that merits additional discussion is the fraction of relaxed versus coherently strained film. While one may not expect a precise quantitative estimate, a reasonable order-of-magnitude assessment of the likely relaxed fraction would strengthen the interpretation. This is particularly relevant because the percolation threshold through optimally doped regions required for superconductivity could be small; providing a plausible range would help readers better evaluate the physical picture.

We thank the referee for this helpful suggestion. We performed a simple intensity analysis to estimate the fraction of coherently strained versus partially relaxed regions for superconducting $\text{La}_3\text{Ni}_2\text{O}_7/\text{SLAO}$ (Fig. R1). Because the estimate depends on the metric used, we considered two approaches. Using the peak heights of the corresponding diffraction features yields an estimate that approximately 78% of the film remains coherently strained, while using the integrated peak areas gives a smaller value of about 18%. These two approaches therefore suggest a plausible range for the coherently strained fraction of roughly 20–80% in the superconducting film. While this estimate should be regarded as an order-of-magnitude estimation, it indicates that a substantial portion of the film is coherently strained, consistent with the scenario discussed in the manuscript in which superconductivity emerges in pristine crystallographic regions.

Figure R1 | Reciprocal space map (RSM) analysis of strain relaxation in superconducting $\text{La}_3\text{Ni}_2\text{O}_7/\text{SLAO}$ sample (Sample A1 from E. K. Ko et al. *Nature* 638, 935-940 (2025))

(a) RSM around the film reflection. The red rectangle indicates the integration window used to extract the line profile along the H direction. (b) Line profile of the scattered intensity obtained from the region marked in (a). The profile can be decomposed into a sharp central peak corresponding to the coherently strained film (red) and a broader component associated with partially relaxed regions (blue). The green symbols represent the experimental data and the black curve the overall fit.

We have also conducted similar RSM analysis for films grown on NGO and STO for completeness (Fig. R2). Peak height metric reveals roughly 95% of film is strained to NGO and 93% to STO substrates, while the integrated peak area metric estimates smaller values of 50% and 37% for NGO and STO, respectively. In both cases, the relaxed fraction is smaller than in the superconducting film on SLAO with the highest strain. Extensive LAO substrate twinning causing multiple sample peaks makes equivalent analysis challenging for films on LAO, but we expect the fraction of coherently strained sample to fall between NGO and STO parameters given the moderate strain state of -0.9% on LAO.

Figure R2 | Reciprocal space map (RSM) analysis of strain relaxation in sample $\text{La}_3\text{Ni}_2\text{O}_7$ grown on NGO (a,b) and STO (c,d) based on data from Supplemental Fig. S9. RSM of $\text{La}_3\text{Ni}_2\text{O}_7$ thin films with the pseudo-tetragonal (1017) Bragg peaks of $\text{La}_3\text{Ni}_2\text{O}_7$ alongside (332) peak of NGO (110) (a), and (103) peak of STO (001) (c). The red rectangle indicates the integration window used to extract the line profile along the Q_x direction for NGO (b) and STO (d).

We added the above discussion of RSM and our estimates of coherently strained fractions to the Supplemental Information Sec. V:

“We performed a simple intensity analysis to estimate the fraction of coherently strained versus partially relaxed regions for superconducting $\text{La}_3\text{Ni}_2\text{O}_7$ /SLAO using RSM given in [E. K. Ko et al. *Nature* 638, 935-940 (2025)]. Because the estimate depends on the metric used, we considered two approaches. Using the peak heights of the corresponding diffraction features yields an estimate that approximately 78% of the film remains coherently strained, while using the integrated peak areas gives a smaller value of about 18%. These two approaches therefore suggest a plausible range for the coherently strained fraction of roughly 20–80% in the superconducting film. While this estimate should be regarded as an order-of-magnitude estimation, it indicates that a substantial portion of the film is coherently strained. Similar analysis performed on RSM data provided in Supplemental Fig. S9 yields a smaller relaxation fraction for films on NGO and STO.”

Regarding the interpretation of the X-ray diffraction (XRD) data mentioned in the rebuttal, the authors could refine their language and analysis. XRD does not simply “average” atomic positions in a volumetric sense; rather, scattering amplitudes from distinct regions combine coherently or

incoherently, both of which deliver intensities characteristic of phases present in the sample (such as the superlattice peaks seen in Fig. 2e). Clarifying this point—and its implications for strain heterogeneity—would improve the rigor of the structural argument.

We thank the referee for pointing out areas requiring further clarification. Along with the analysis presented above, we have further added clarification for the interpretation of RSM data to the Supplemental Information Sec. V:

“As shown in Extended Data Fig. E6, the thin films involve areas with relaxation and defects that inherently scatter differently compared to the pristine regions of the film. Therefore, it is crucial to account for the various constructive and destructive interference occurring due to defects as well as the pristine bilayer through extensive simulation for quantitative interpretation of RSM data (Supplemental Fig. S9). Here, we present a rough estimate, not a quantitative one, of the strain heterogeneity present in the thin films through RSM analysis.”

The question of whether the coherently strained fraction of the film is intrinsically superconducting is central to the manuscript. Relevant recent ARPES work (arXiv:2502.17831) the authors could include in a revised manuscript suggests a superconducting gap opening below T_c . At the same time, other recent studies (e.g., arXiv:2504.16372v2) highlight open questions about the electronic structure and the nature of the gap above T_c . Framing the manuscript in this broader and still-evolving context—acknowledging both supportive and critical evidence—would make the claims more balanced and persuasive.

We agree with the reviewer in putting forth a balanced ensemble of relevant, well-supported claims. As detailed above, we have expanded our discussion regarding efforts to estimate the coherently strained fraction, and more explicitly acknowledged the possibility of filamentary superconductivity:

“While there has been mounting evidence to suggest bulk-like superconductivity hosted in the pristine bilayer regions of the thin films [Y. Liu et al. *Nature Material* 24, 1221–1227 (2025)], we cannot rule out the possibility of filamentary superconductivity in the defect regions.”

We have also added a reference to two ARPES papers mentioned by the reviewer emphasizing the need for precision structural measurements to complement the electronic structure and gap evolution discussed in each of those works:

“These considerations are particularly relevant for the interpretation of recent photoemission experiments [B. Y. Wang et al. *arXiv:2504.16372* (2025), J. Shen et al. *arXiv:2502.17831* (2025)], where discrepancies in band positions could arise from structural relaxation near defects or at the film surfaces.”

The authors are commended for including the Hall data, which adds relevant information about charge transport and doping. This is a welcome addition. However, the manuscript’s use of these data to argue for close similarity between PLD- and MBE-grown films is an overly strong claim.

Given known differences between these growth techniques, a more nuanced discussion of similarities and potential differences would be appropriate.

As noted above, we have expanded the discussion of different growth methods in relation to the film properties in the manuscript. We hope our detailed work here provides a practical roadmap for other groups to perform a direct comparison of different growth methods in the future.

Once superconductivity is demonstrated in the MBE-grown films, this becomes a promising contribution to the field. Additionally, with targeted revisions that clarify the strained versus relaxed fractions, refine the XRD interpretation, and more carefully situate the superconductivity claim within the current literature, the manuscript has the potential to make a meaningful impact.